# Federated Learning in Edge Computing: A Systematic Survey

**DOI:** 10.3390/s22020450

**Published:** 2022-01-07

**Authors:** Haftay Gebreslasie Abreha, Mohammad Hayajneh, Mohamed Adel Serhani

**Affiliations:** Department of Computer and Network Engineering, College of Information Technology, United Arab Emirates University, Al Ain P.O. Box 15551, United Arab Emirates; 202090183@uaeu.ac.ae (H.G.A.); mhayajneh@uaeu.ac.ae (M.H.)

**Keywords:** federated learning, edge computing, intelligent edge, edge AI, data privacy, data security

## Abstract

Edge Computing (EC) is a new architecture that extends Cloud Computing (CC) services closer to data sources. EC combined with Deep Learning (DL) is a promising technology and is widely used in several applications. However, in conventional DL architectures with EC enabled, data producers must frequently send and share data with third parties, edge or cloud servers, to train their models. This architecture is often impractical due to the high bandwidth requirements, legalization, and privacy vulnerabilities. The Federated Learning (FL) concept has recently emerged as a promising solution for mitigating the problems of unwanted bandwidth loss, data privacy, and legalization. FL can co-train models across distributed clients, such as mobile phones, automobiles, hospitals, and more, through a centralized server, while maintaining data localization. FL can therefore be viewed as a stimulating factor in the EC paradigm as it enables collaborative learning and model optimization. Although the existing surveys have taken into account applications of FL in EC environments, there has not been any systematic survey discussing FL implementation and challenges in the EC paradigm. This paper aims to provide a systematic survey of the literature on the implementation of FL in EC environments with a taxonomy to identify advanced solutions and other open problems. In this survey, we review the fundamentals of EC and FL, then we review the existing related works in FL in EC. Furthermore, we describe the protocols, architecture, framework, and hardware requirements for FL implementation in the EC environment. Moreover, we discuss the applications, challenges, and related existing solutions in the edge FL. Finally, we detail two relevant case studies of applying FL in EC, and we identify open issues and potential directions for future research. We believe this survey will help researchers better understand the connection between FL and EC enabling technologies and concepts.

## 1. Introduction

According to Cisco, the number of connected IoT devices could exceed 75 billion by 2025, which is 2.5-times the amount of data produced in 2020 (i.e., 31 billion) [1]. Furthermore, IoT devices are equipped with heterogeneous and advanced sensors for various crowdsensing applications such as smart industry [2], healthcare [3], and Unmanned Areal Vehicle (UAV) [4] applications. In addition, the demand for time- and quality-sensitive IoT applications is overwhelming currently, which requires an infrastructure with high availability and resilience. However, managing massive, heterogeneous, and distributed IoT data and providing services at a specified performance with cloud infrastructure looks impossible. Edge Computing (EC) is a new architecture that extends Cloud Computing (CC) services closer to data sources, which reduces the latency and bandwidth cost and improves the resilience and availability of the network [5,6,7]. Thus, time-critical applications with specified Service Level Agreement (SLA) demands can be fulfilled by leveraging the EC-enabled architecture. In addition, EC is a distributed computing paradigm that can handle the proliferation of IoT data and take advantage of distributed heterogeneous computing resources. EC combined with Deep Learning (DL) [8] is a promising technology and is widely used in several applications.

Data storage and model training occur on high-performance cloud servers in conventional centralized ML approaches. Multiple edge nodes also collaborate with the remote cloud to perform large distributed tasks that include both local processing and remote coordination and execution. However, due to the inherent challenges listed below, transmitting all data collected from edge devices to a central data center for training a model over the network is not feasible:Communication cost: Sending large amounts of data from EC nodes or edge devices to a remote server requires further network traffic encoding and transmission time. In other words, insufficient bandwidth negatively affects the efficiency of data transmission. In addition, cloud servers are often far from end-users, where data need to travel across multiple edge nodes. Therefore, a network with thousands of edge devices lacks the ability to meet the real-time, low-latency, and high Quality of Service (QoS) requirements due to the long-distance data transmission [9]. Therefore, the traditional cloud-based architecture is not suitable to accomplish the requirements mentioned above;Reliability: Clients send their datasets via different communication network connections to the remotely located cloud server in conventional centralized model-training architectures. Therefore, the wireless communications and core network connections between clients and servers affect DL model training and inferences to a large extent. Hence, the connection has to be reasonably reliable even when there is an interruption in the network. Nevertheless, a centralized architecture faces system performance degradation and possible failure because of the unreliable wireless connection between the client and server, which can significantly affect the model;Data privacy and security concerns: Due to concerns about privacy and unauthorized access to their data, users are often hesitant to share their information [10]. As a result, the specific implementation of a set of controls, applications, and techniques that identify the relative importance of various datasets, their sensitivity, compliance requirements, and the application of appropriate safeguards to secure these resources is required. Traditional centralized training, however, is vulnerable to sensitive data privacy breaches, intruders, hackers, and sniffers since clients have to share their raw data with third parties, such as cloud or edge servers, to train a model;Administrative policies: Data owners are becoming more concerned about their privacy. Following public fears about privacy in the age of Big Data, legislators have responded by enacting data privacy legislation. For example, the General Data Protection Regulation (GDPR in the European Union) [11], California Consumer Privacy Act (CCPA) [12] in the USA, and the Personal Data Protection Act (PDPA) in Singapore [13] intend to restrict the collection of data to only those that are needed for processing and consented to by consumers. Privacy legalization cannot be achieved by the traditional centralized model-training architecture since clients must send raw data to the server for model training.

Federated Learning (FL) is a concept developed by Google researchers in 2016, as a promising solution for addressing the issues of communication costs, data privacy, and legalization [14,15,16,17,18,19]. An FL approach is a distributed ML approach where models are trained on end devices, organizations, or individuals under centralized control without sharing their local datasets. This ensures the privacy of data during the training process. An edge server or cloud server periodically gathers the trained parameters to create and update a better and more accurate model, which is sent back to the edge devices for local training. Generally, there are five steps in the FL training process. The FL server first determines an ML model to be trained on the clients’ local database. Second, a subset of current clients is chosen at random or using client selection algorithms such as Federated Client Selection (FedCS) [20]. Third, the server multicasts the initial or updated global model to the selected clients. The clients download the current global model parameters and train the model locally. Then, each client in the subset sends updates to the server in the fourth step. Finally, the FL server receives the updates and aggregates them using aggregation algorithms such as FedAvg [16] to generate a new global model without accessing any clients’ data. The FL server orchestrates the training process and transmits the global model updates to the selected clients each round. The steps iterate until the desired level of accuracy is achieved.

FL has several distinct advantages over traditional centralized ML training. The time and bandwidth required for training and inference are significantly degraded because local data are used and not frequently sent to a remote server. Thus, the updated model can be used for prediction on the user’s device, for which FL ensures user privacy and security as the data remain on the personal device. Moreover, collaborative learning using FL is easy and consumes less power as the models are trained on edge devices. The term implies that edge computing is a suitable environment for using FL. It is a technology that enables the training of ML models on mobile edge networks. Therefore, the communication costs, security, privacy, and legalization issues could be alleviated by leveraging FL in the EC paradigm. Figure 1 shows how FL works in the context of edge computing. Depending on how the global learning model is implemented, there are three types of FL structures [21]: cloud-enabled, edge-enabled, and hierarchical (client-edge-cloud-enabled) FL. Edge-enabled FL includes a group of devices in close proximity, so a global learning model can be computed on the edge server. For aggregation, local models, after being trained locally, are then sent to the edge server close to the edge devices. An edge server aggregates and updates the model and then broadcasts it to the end devices. In contrast, a cloud-enabled FL model computes a global learning model for edge systems that are geographically distributed over a vast range of areas. Clients within a client range of the edge server will collaborate on DL model training. The edge server will be the parameter server for edge-enabled FL. In contrast to a server residing on the cloud, the parameter server is typically located near the end-user, thus reducing the communication latency. However, the edge servers are often resource constrained, which limits their computational efficiency. The total number of clients participating in cloud-based FL will exceed millions, resulting in big datasets for DL. As a result of network congestion, the connection with the cloud server is slow and unpredictable, resulting in an inefficient training process. The authors of [15] described the tradeoff between communication efficiency and the aggregation convergence rate, such as more local computation utilized at the cost of less communication. The parameter server, on the other hand, is located in the closest location in the edge network, such as a base station. As a result, the edge parameter server’s computation latency equals the contact latency. As a result, improved computation and networking tradeoffs are possible. However, one disadvantage of edge-based FL is that each server can only support a limited number of clients, resulting in the degradation of training performance over time.

On the right side of Figure 1, FL with a hierarchical structure is illustrated, which makes use of a cloud server to access the enormous training samples and use its local clients to update the model quickly. By employing hierarchical FL, cloud communications will be significantly reduced, complemented by efficient client–edge updates. Due to this, the training runtimes and iterations will decrease significantly as compared to cloud-based FL. The term “client” is used in this article to refer to devices or nodes that undertake local ML training to generate the global FL model. As a result, the clients are determined by the FL settings. The use of FL, originally designed for mobile devices, has expanded rapidly into many other applications, including collaborating to train a model among many organizations. “Cross-device” and “cross-silo” are terms used to describe the possible FL settings for end devices and organizations as clients correspondingly [18]. Although the existing surveys have taken into account applications of FL in EC environments, there has not been any systematic review discussing FL implementation and challenges in the EC paradigm. The purpose of this paper is to provide a systematic literature review of FL deployment in EC environments, with a taxonomy that identifies advanced solutions and other open issues. In this survey, we begin by discussing the fundamentals of EC and FL. Then, we review the existing related works in FL in EC. Furthermore, we described the protocols, architecture, framework, and hardware requirements for FL implementation in the EC environment. Moreover, we discuss the applications, challenges, and related existing solutions in the edge FL. Finally, in addition to pertinent case studies, we identify open issues and potential directions for future research. Finally, our goal in conducting this survey is to provide readers with a better understanding of how FL and EC technologies and concepts are linked.

As for the remainder of the paper, Section 2 describes EC, DL, and FL. Section 3 summarizes the literature review, which includes relevant review articles and explains what this paper contributes. The methods of research adopted in this study are described in Section 4. Section 5 discusses the architecture, challenges, and state-of-the-art solutions for FL-enabled EC. Section 6 describes case studies involving UAVs and healthcare. Section 7 explains the open issues and possible future research directions. Section 8 concludes the paper.

## 2. Fundamentals of Edge Computing and Federated Learning

Before diving into the results of our systematic survey, it is important to understand the background and fundamentals of both EC and FL. In this section, we first provide this overview, and then, we provide an overview of recent FL and edge computing surveys.

### 2.1. Edge Computing

Computing is becoming more ubiquitous, and services are overflowing from the cloud to the edge. To solve this problem, there is a range of devices from cloud-based servers to smartphones and wearables to IoT devices. As a result, Big Data data sources are shifting from large-scale data centers on the cloud to diversified dispersed data sources with advanced computing capabilities and enormous edge devices. However, for a variety of reasons, the current cloud-based computing paradigm appears to be increasingly incapable of managing and analyzing the collected/produced data at the edge [22]. Big Data generated at the edge must be routed to a cloud server for processing, which is typically located far away from the end device. This scenario will not be practical for time-sensitive applications such as augmented reality, virtual reality [23], and autonomous vehicle network systems [24]. Furthermore, with the growth of data quality generated at the edge, network bandwidth between the edge devices and server is one of the bottlenecks for the cloud-based computing paradigm. In addition to computational and communication costs, IoT devices need to send raw data to the cloud server for processing. These include sending sensitive data to the server, such as patient information, which not only violates privacy, but also threatens data security due to frequent data transmission [25].

Therefore, it would be more efficient to process the data at the edge of the network described as edge computing. There are many new concepts aiming to operate on the edge of the network in the evolution of EC, including Micro Data Centers (MDCs) [26], Cloudlets [27], mobile edge computing [28] (also called multi-access edge computing [29]), and fog computing [30,31]. However, there has not yet been agreement within the EC community regarding the standard definitions, architecture, and protocols for EC [32]. This group of upcoming technologies is commonly referred to as “edge computing”. Edge computing brings compute power and data storage closer to the edge devices through the use of distributed computing. This technology is emerging as an essential tool to mitigate the bottleneck of emerging technologies. It reduces data transmission, improves service latency, eases cloud computing pressure by leveraging distributed computing, and enhances security and privacy. The concept of EC does not exclude the cloud computing paradigm [33,34]. Edge computing, on the other hand, is a supplement to and extension of cloud computing. Edge computing combined with cloud computing has three advantages over cloud computing alone [25]: (1) the distribution of edge computing nodes can handle many computation tasks without requiring the data to be exchanged with the cloud, thus decreasing traffic on the backbone network; (2) data transmission delays can be reduced and response times can be improved by hosting services at the edge; (3) the backup to the Cloud is powerful because the cloud can process many computations without exchanging data, thereby reducing the amount of network traffic. EC does not simply copy and transmit cloud capabilities to the edge since the edge differs from the cloud server in functionality. The edge is small, resource-constrained, and heterogeneous. Furthermore, edge nodes are dispersed towards the network’s far end.

One may ask how edge computing delivers services and applications compared to cloud computing. One of the main differences between edge and cloud computing is that edge devices are only data consumers in the cloud computing paradigm, whereas they are both data consumers and producers in EC, as described in Figure 2. Different types of mobile devices and sensors, such as the IoT, Big Data, and social platforms connected to the core network via the edge network, can be data producers or consumers, as shown in Figure 2. The core network is connected to either or both private and public cloud networks. The evolution of mobile networks, in particular 5G and beyond, brings cloud services near the edge devices. Figure 2a shows the conventional cloud computing structure. Data producers generate raw data and transfer them to the cloud, and data consumers send a request for consuming data to the cloud, as noted by the solid line (1). The dotted line (2) indicates the request for consuming data being sent from data consumers to the cloud, and the result from the cloud is represented by the dotted line (3) where data are distributed over the data consumers. This structure, however, is insufficient for the IoT. First, the volume of data at the edge is excessive, resulting in a massive waste of bandwidth and computing resources. Second, the demand for data privacy will be an impediment to cloud computing in the IoT. Finally, because most IoT end nodes are energy constrained and the wireless communication module is typically quite energy hungry, shifting some processing operations to the edge could saves energy.

In the edge computing architecture depicted in Figure 2b, the computing distribution platform, for example AlibabaCloud [35], which includes applications such as data caching/storage, computational offloading, data processing, request distribution, service delivery, IoT management, security, and privacy detections, efficiently and effectively performs on the edge network to achieve low latency, large bandwidth, and the mass connection of the services. It is easier to manage the end-to-end connection and resource sharing on the platform. As a result of the edge-based infrastructure, data can be produced and consumed in close proximity to the edge itself, which reduces the need for data to be moved back to the center in order to meet requirements such as security, reliability, and privacy.

### 2.2. Deep Learning

DL has attracted the attention of all researchers, academicians, and industry personnel working in a variety of fields, including Computer Vision (CV) [36,37], healthcare [38], game playing [39,40], autonomous vehicles [41,42], and Natural Language Processing (NLP) [43,44] without the costly hand-crafted feature engineering required in conventional ML. DL is a subfield of ML concerned with algorithms inspired by the structure and function of the brain called Artificial Neural Networks (ANN). Furthermore, a Deep Neural Network (DNN) is an ANN with multiple layers between the input and output layers. To automatically identify and learn automatic feature extraction from Big Data datasets, DL employs Neural Networks (NNs) [45,46]. Then, it uses these features in later steps to classify the input, make decisions, or generate new information [47]. Neural networks are modeled after the brain and consist of multiple layers of logistic regression units called neurons. Neural networks are known to be able to learn complex hypotheses for regression and classification. Conversely, training neural networks is difficult, as their cost functions have many local minima. Hence, the training tends to converge to a local minimum, resulting in the poor generalization of the network. For the last ten years, neural networks have been celebrating a comeback under the term deep learning, taking advantage of many hidden layers to build more powerful ML algorithms. Feed-forward and backpropagation algorithms are the backbones of neural network architectures.

As the simplest neural network, feed-forward networks have three layers: an input layer, one or more hidden layers, and a final output layer. The input is weighted and biased through a non-linear activation function to produce the output in the conventional feed-forward neural network. Furthermore, the choice of the activation function in the hidden layer is critical to design an effective DNN model that will control how well the network model learns the training dataset. Sigmoid, Softmax, and Rectified Linear Unit (ReLU) are some of the desired activation functions used by DNNs. Table 1 shows the mathematical expressions of the activation function. There are often many hidden layers in a DNN that map the input to the output. For example, a classification DNN produces a vector of scores for an image input based on the positional index of the highest score. Multi-layered DNNs minimize the deviation between the actual value and the output, also known as the loss function. Let us consider N number of input features designated as $x_{1}$, $x_{}$, ..., $x_{N}$, with weights of $w_{1}$, $w_{}$, ..., $w_{N}$, respectively. f(·) is an activation function. Therefore, the output, *y*, is driven from the individual weights and input feature, as illustrated in Equation (Equation 1).
(1)$$\begin{array}{c} y=f(\sum_{n=1}^{N}x_{i}*w_{i}+b) \end{array}$$

To minimize loss functions such as the Mean-Squared Error (MSE), Cross-Entropy (CE), and Mean Absolute Error (MAE), DNNs use backward propagation techniques. Back-propagation is the core of NN training. It is a technique for fine-tuning the weights of a neural network based on the previous epoch’s error rate (i.e., iteration). By fine-tuning the weights, you may lower the error rates and improve the model’s generalization, making it more dependable. Optimizing techniques such as Stochastic Gradient Descent (SGD) are used to calibrate the network weights. Weight updates are obtained by multiplying gamma by the partial derivative of the loss function *L* with respect to the weight *W*. The gradient descent iteration step size is the gradient descent learning rate. SGD is evaluated using the following formula:(2)$$\begin{array}{c} W=W-\gamma \frac{\partial L}{\partial W} \end{array}$$(3)$$\begin{array}{c} \frac{\partial L}{\partial W}\approx \frac{1}{m}\sum_{i\in B}\frac{\partial l^{\left(i\right)}}{\partial W} \end{array}$$

It should be noted that, in (2), the SGD formula is a minibatch GD formula. The gradient matrix in Equation (Equation 3) is calculated as an average of the gradient matrices over all B batches, where there are *m* training samples in each batch. As a result, the partial derivative is preferable to full-batch GD, which employs the entire training set. Full-batch GD can cause training and batch memorization to be delayed [48]. Back-propagation is used to deduce gradient matrices from the gradient error input *e* described [49,50] as $e=\frac{\partial L^{B}}{\partial y}$. To reduce the cost, the training iterations of both feed-forward and backward propagations are repeated over multiple epochs. Properly trained DNNs generalize well, achieving high inference accuracy when they are applied to a dataset they have not encountered before, such as the test set. DNN learning can not only be supervised, but also semi-supervised [51], unsupervised [52], and use reinforcement [53] learning models. There are different types such as Convolutional Neural Networks (CNNs) [54], Recurrent Neural Networks (RNNs) [55], and Multilayer Perceptrons (MLP) [56], which have already been applied to different themes.

### 2.3. Enabling Deep Learning at the Edge

Currently, DL training data are pre-processed in the proximity of the edge devices prior to the cloud transmission. Due to the constant transmission of massive data from the clients to the cloud, this approach consumes both many computational and communication resources, thus preventing further algorithm performance improvement. Furthermore, the client to cloud training architecture is not feasible for DL services requiring locality and continuous training. All of these issues highlight the demand for a new training system to supplement the current training scheme in the cloud. “DL training at the edge” refers to model training that takes place at the edge, or perhaps among “end–edge–cloud” nodes, with the edge serving as the core architecture [25]. Distributed DNN models are well adapted to EC. To collectively train the DL model, the workload is dispersed among numerous network computing centers such as the cloud, base stations, edge nodes, and end devices by assigning each a small portion of the work to perform and then aggregating the results. Subtasks for training are allocated based on the edge device and distributed models that prioritize load balancing in inference tasks. The training process can be made more efficient by using parallel servers. Data parallelism and model parallelism are the two most frequent approaches to distributed training. Model parallelism divides a big DL model into several sections, then feeds data samples to these segmented models for parallel training. Model parallelism improves both the training speed and solves the problem of having a large model, which is beyond the device memory size. Training of the DL model, in general, demands a considerable amount of computing resources, such as thousands of CPUs. Therefore, distributed GPUs are often used for parallel training to solve the challenge. As an alternative to data parallelism, data partitioning involves building a model from copies of the data and training them simultaneously on their data samples, which increases training efficiency dramatically. However, most distributed DNN architectures cannot handle heterogeneous datasets such as non-IID and unbalanced data while also dealing with legalization and privacy concerns. Furthermore, the overall system performance degrades when they deal with the heterogeneous environment. FL has earned the attention of by many researchers and academicians to resolve these difficulties appearing in centralized DL training.

### 2.4. Federated Learning

In 2016, McMahan [16] first coined the term federated learning to address privacy concerns. As described in the Introduction, FL sets up many distributed clients, collaboratively training a model by leveraging decentralized local datasets with the support of a centralized server that controls and monitors the operation. Clients’ raw data remain local and share the parameter updates with the server to achieve the required model performance [18]. FL also enhances the data collection and diminishes the costs and risks associated with centralized ML. Furthermore, edge computing is enabled by FL because it allows DL models to be trained collaboratively to optimize networks.

The application of FL techniques on edge networks has the following advantages over the traditional centralized ML model. Data owners send update parameters instead of raw data to the FL server, which lowers the number and size of the communication data. Therefore, it enhances network bandwidth utilization [57]. Second is the latency: in time-critical applications [58] (i.e., industrial control, mobility automation, remote control, and real-time media). Furthermore, applications that require real-time decisions, such as event detection, augmented reality, and medical applications, can be processed locally at end devices to improve performance [59]. As a result, FL systems have substantially lower latency than centralized systems. Third is privacy: raw data are not sent to a central server, ensuring the privacy of each user. More users can therefore train a better model if the raw data are not sent to a central server [60]. FL enables participating clients to cooperatively train a global model utilizing their joined data without disclosing each device’s personal information to the centralized server. The three-step approach shown in Figure 3 is utilized to achieve this privacy-preserving collaborative learning technique [61]:Task initialization: In a specific interval, the server selects a certain number of devices from the thousands available. It determines the target application and data requirements once the training task is specified. In addition, the server sets the hyperparameters related to the model and training process, such as the learning rate. Specifically, it initializes the weights on the server by leveraging weight initialization methods such as random, He, or Xavier initialization [3]. The parameter server disseminates the global model $w_{G}^{0}$ and the FL task to the selected participants after specifying the devices;Local model training: The participants receive the global model $w_{G}^{t}$, where *t* denotes the current iteration index, and each participant updates the local model parameters $w_{i}^{t}$ based on their local data and device. The objective of client *i* is therefore to obtain an optimal parameter $w_{i}^{t}$ at the *t* time iteration at the minimum value of the loss function $L\left(w_{i}^{t}\right)$ [50]:
$$\begin{array}{c} w_{i}^{t}=\arg\min_{w_{i}^{t}}\;L\left(w_{i}^{t}\right) \end{array}$$Finally, each local model’s updated parameters are sent again back to the FL parameter server;Global model aggregation: The centralized server receives the local parameters from each participant and aggregates the local models from the participants, then sends the updated global model parameters $w_{G}^{t+1}$ back to all the participating clients to minimize the global loss function, $L\left(w_{G}^{t}\right)$, i.e.,
$$\begin{array}{c} L\left(w_{G}^{t}\right)=\frac{1}{N}\sum_{n=1}^{N}L\left(w_{i}^{t}\right) \end{array}$$

Furthermore, Steps 2 and 3 are repeated in the iteration process until the global loss function achieves the optimal accuracy.

## 3. Literature Review

This section presents the previous surveys conducted by researchers in FL in the EC paradigm. The survey helps readers to comprehend the difference between previous surveys and the one reported in this study. Furthermore, the contribution of this paper is presented in this section briefly.

### 3.1. Related Works on Federated Learning in Edge Computing

To the best of our knowledge, there is no other systematic survey of work on FL in EC. Despite the fact that there are surveys on both EC and FL, most studies handle the two areas independently. Furthermore, most of the literature has not considered the hardware requirement challenges. The authors of [7] presented a survey on architecture and computational offloading in MEC explicitly. The convergence of the intelligent edge and edge intelligence was studied in [25]. Moreover, Nguyen et al. [62] analyzed DRL solutions to address emerging issues in communications and networking. In [33], the authors provided a tutorial on fog computing and its related computing paradigms, challenges, and future directions. Similarly, the authors of [6] described how mobile computing and wireless communication resources are managed together in MEC. Furthermore, in [63], the authors studied the architectures and frameworks for edge intelligence. The authors of [64] analyzed edge intelligence specifically for 6G networks. In addition, Cui et al. [65] reviewed the applications of ML for IoT management. Similarly, in [66], they presented a survey on computation offloading approaches in mobile edge computing. In addition, the authors of [67] investigated the techniques for computational offloading. Abbas et al. [68] studied the architectures and applications of MEC. Furthermore, in [69], the authors reviewed the computation offloading modeling for EC. In addition, the authors of [70] investigated the computing, caching, and communication issues in MEC. Furthermore, Yao et al. [71] analyzed the phases of caching and the differences between caching schemes. Moreover, in [72], they provided a survey on MEC for the 5G network architecture. However, none of the works [25,33] and [64,66,69,70,71,72,73,74,75,76,77,78,79] considered the application of FL. Furthermore, all the literature reviews except [12,71] are not SLRs. In addition, the surveys did not consider the hardware requirement challenges, except [25].

Mothukuri et al. [80] provided a study focusing on the data privacy and security of FL viewpoints and described the areas that require in-depth research. FL applications from the industrial engineering and computer science perspectives were investigated by Li et al. [81]. Furthermore, asynchronous training, gradient aggregation, returned model verification, block-chain-based FL, and federated training for unsupervised ML of the six research fronts they outlined in their definition of FL. In addition, the authors emphasized outstanding research concerns and obstacles in future study fields that may be optimized. In addition, Zhang et al. [61] discussed existing FL research from five perspectives: data partition, privacy techniques, relevant ML models, communication architecture, and heterogeneity solutions. They also noted the challenges and potential research directions.

Furthermore, Li et al. [82] proposed an FL building block taxonomy that classifies the FL system into six different aspects: data distribution, ML model, privacy mechanism, communication architecture, the scale of the federation, and the motivation for FL. The authors presented the design factors, case studies, and future research opportunities. Similarly, in [19], the authors provided a brief tutorial on FL and the challenges of FL, offered a broad overview of the literature, and highlighted several future research directions. In particular, they considered four challenges of FL: communication efficiency, systems heterogeneity, statistical heterogeneity, and privacy. In addition, Kairouz et al. [18] provided a broad overview of current research trends and relevant challenges raised by researchers. The authors focused on communication efficiency, data heterogeneity, privacy, and model aggregation as particular challenges. The authors of [83] discussed the opportunities and challenges in FL. The authors of [84] described the existing FL and proposed the architecture of FL systems. Another focus was on describing the architecture and classification of the different FL configurations used for different training data distributions, as described in [17]. The authors of [85] described the applicability of FL in smart city sensing, as well as clear insights on open issues, challenges, and opportunities. Moreover, Lyu et al. [86] presented a survey paper focusing on the security threats and vulnerability challenges in FL systems; furthermore, the authors of [87] summarized the most used defense strategies in FL. In addition, the authors concluded that employing a single defensive method is insufficient to provide adequate security against all possible attack modes. Similarly, Reference [88] provided protocols and platforms to develop better privacy solutions for industries that desperately need them. Moreover, Lo et al. [89] performed an SLR on FL from the software engineering perspective.

However, the authors of [17,18,19,61,80,81,82,83,84,85,86,87,88,89] focused only on the application of FL and did not explore the impact of the hardware requirements. Furthermore, they did not include the implementation challenges of FL in the EC paradigm. In addition, except in the articles [82,89], they did not adopt the SLR methodology for the surveys. The authors of [90] highlighted FL’s applications in wireless communications, specifically the cellular network architecture. They did not address the challenges of FL implementation in edge networks. In addition, the article [50] discussed the basics and problems of FL for edge networks, as well as prospective future research approaches. Moreover, the authors of [91] described the FL implementation challenges, methods, and future directions in 6G communications. Khan et al. [92] also discussed FL’s recent advancements in enabling FL-powered IoT applications to run on IoT networks. They highlighted a number of open research challenges, as well as potential solutions. In addition, Reference [93] provided a comprehensive review of combining FL with the IoT in terms of privacy, resource management, and data management, as well as the problems, potential solutions, and future research directions. Similarly, the authors of [94] highlighted the requirements and obstacles of FL implementation in wireless communications, particularly for 6G wireless networks. Moreover, Reference [95] investigated and analyzed FL’s potential for enabling a wide range of IoT services, such as IoT data sharing, data offloading and caching, attack detection, localization, mobile crowdsensing, and IoT privacy and security, as well as the current challenges and possible directions for future research in the field.

The authors of [96] provided a thorough overview of FDL applications for UAV networks, as well as technical obstacles, unresolved questions, and future research objectives. Moreover, the authors of [97] discussed the implementation challenges of FL techniques for the full integration of the communication, computation, and storage capabilities of F-RANs. Moreover, future trends of FL-enabled intelligent F-RANs, such as potential applications and open issues, were discussed. In [98], they reviewed FL applications and the advances of FL towards enabling the vehicular IoT. Finally, the authors provided a few open research challenges regarding the FL-enabled vehicular IoT. Similarly, the author of [99] explored the key factors responsible for this problem and explored how FL could provide solutions for the future of digital health and the challenges and trends in implementing FL. In addition, Reference [100] covered the applications of FL to autonomous robots and introduced the key background concepts and considerations in current research. The authors of [101] provided a review of FL technologies within the biomedical space and the challenges. Moreover, the authors of [102] presented a comprehensive tutorial on FL in the domain of communication and networking. However, none of the publications in [50,90,91,92,93,94,95,96,97,98,99,100,101,102] adopted the SLR methodology. Moreover, the authors of [50,90,91,94,102] did not take into account the challenges of FL implementation in the EC paradigm. Furthermore, the existing literature except [7,94] did not analyze the hardware requirements in implementing FL in the EC environment.

Table 2 summarizes the comparison of our survey with current surveys in the literature based on a set of relevant criteria including the Systematic Literature Review (SLR) methodology, Edge Computing (EC), Federated Learning (FL), Case Studies (CS), Hardware Requirements (HWR), and Future Research Directions (FRD). First, we analyzed if the existing related survey papers used an SLR approach to identify the challenges and directions for FL implementation within the EC paradigm. For their respective applications in the domain of federated learning, the authors of [62,66,82,89] applied different SLR methodologies. The SLR methodology, however, has never been used in most related surveys. In our screening, almost all of the related survey papers addressed the open research issues and directions, except for the papers by [65,72,83,87,88]. In addition, we examined whether survey papers described the results of their research supported by case studies as a means of explaining the features of the field. Therefore, most of the literature did not show results illustrated by use-cases, as can be seen in the table. Furthermore, we analyzed the review articles to determine if they described FL and EC clearly in such a way as to characterize the challenges and future research trends. As we can see in the table, there are surveys on EC and FL, but the existing studies treated these two topics separately. In [6,7,25,33,62,63,64,65,66,68,69,70,71,72], the authors explored edge computing without including federated learning. Similarly, the authors of [17,18,19,61,81,82,83,84,85,86,87,88,90,91,92,93,96] described generic federated learning without relating to edge computing. In addition, we examined whether the current reviews explored and discussed the HWR challenges in their study, which provides the knowledge of hardware limitations and mitigating techniques. A recent article [25,102] identified the hardware constraints in edge computing. However, they were unable to demonstrate the hardware requirements to implement federated learning models in an edge computing environment. As far as we know, the survey papers did not include FL, EC, HWR, and FRD using the SLR methodology that we used. Therefore, we analyzed FL implementation in EC by adopting an SLR research methodology in this paper. Furthermore, we also discussed the HWR and described the possible FRD backed by case studies.

### 3.2. Contributions

Although there have been several surveys studies conducted on federated learning in edge computing [17,18,19,61,80,81,82,83,84,85,86,87,88,89,90,91,92,93,94,95,96,97,98,99,100,101,102], there has not been a systematic review of FL in EC yet. With a taxonomy that identifies advanced solutions, as well as other related open problems, this paper provides a systematic review of the literature on FL implementation in EC environments. To illustrate the future scope of the implementation of FL in the EC paradigm, we performed a Systematic Literature Review (SLR) to compare, analyze, explore, and understand the current attempts and directions. In addition, to the best of our knowledge, this is the first review of the state-of-the-art literature examining the impact of architectural and hardware requirements on FL implementation in an edge computing context. First, we review the fundamentals of EC and FL, then we review the existing works related to FL in EC. In addition, we describe the protocols, architecture, framework, and hardware requirements for FL implementation in the EC environment. Additionally, we discuss the applications, challenges, and existing solutions for edge-enabled FL. Finally, we detail two relevant case studies of applying FL in EC, and we identify open issues and potential directions for future research. We believe this survey will help researchers in the field better understand the connection between FL and EC-enabling technologies and concepts. As a result, the main objectives of this paper are as follows:Classifying and describing the techniques and approaches to understand FL in the EC paradigm. Furthermore, this survey will help readers understand the current literature’s tendency and emphasis in the field of EC and FL;Categorizing and analyzing the challenges and constraints in EC to FL implementation settings;Identifying and evaluating state-of-the-art technical solutions to mitigate the challenges in FL implementation in the EC context;Providing case studies that leverage the enabling technologies of the FL and EC paradigm, such as healthcare and autonomous vehicles;Enhancing the understanding of FL implementation in the EC paradigm by providing insights into existing mechanisms and future research directions.

**Table 2 sensors-22-00450-t002:** Comparison of our survey to current state-of-the-art surveys.

References	Summary	SLR	EC	FL	CS	HWR	FRD
[7]	Survey on architecture and computation offloading in MEC	X	√	X	√	X	√
[25]	Survey on convergence of the intelligent edge and edge intelligence	X	√	X	√	√	√
[62]	Communication and networking issues being addressed by DRL	√	√	X	X	X	√
[33]	Survey on fog computing and related edge computing paradigms	X	√	X	X	X	√
[6]	Survey on the integrated management of mobile computing and wireless communication resources in MEC	X	√	X	√	X	√
[63]	Edge intelligence architectures and frameworks’ survey	X	√	X	X	X	√
[64]	Survey on edge intelligence specifically for 6G networks	X	√	X	√	X	√
[65]	Management of IoT systems, e.g., network security and management by leveraging ML	X	√	X	√	X	X
[66]	Survey on computation offloading approaches in mobile edge computing	√	√	X	√	X	√
[67]	Survey on techniques for computation offloading	X	√	X	X	X	X
[68]	Survey on architectures and applications of MEC	X	√	X	√	X	√
[69]	Survey on computation offloading modeling for edge computing	X	√	X	X	X	√
[70]	A MEC survey on computing, caching, and communications	X	√	X	√	X	√
[71]	Comparative study of caching phases and caching schemes	X	√	X	X	X	√
[72]	Survey on MEC for the 5G network architecture	X	√	X	X	X	X
[80]	Survey on the data privacy and security of FL	X	X	√	X	X	√
[81]	Survey on FL applications in industrial engineering and computer science	X	X	√	√	X	√
[61]	Survey on FL research from five perspectives: data partition, privacy techniques, relevant ML models, communication architecture, and heterogeneity solutions	X	X	√	X	X	√
[82]	Proposed an FL building block taxonomy with six different aspects: data distribution, ML model, privacy mechanism, communication architecture, the scale of the federation, and the motivation for FL	√	X	√	√	X	√
[19]	Tutorial on FL challenges	X	X	√	X	X	√
[18]	Overview of current research trends and relevant challenges	X	X	√	X	X	√
[83]	Discussed the opportunities and challenges in FL	X	X	√	√	X	X
[84]	described the existing FL and proposed an architecture of FL systems	X	X	√	X	X	√
[17]	Described the different FL settings in more detail, emphasizing their architecture and categorization	X	X	√	X	X	X
[85]	Discussion on the applicability of FL in smart city sensing	X	X	√	√	X	√
[86]	Survey on the security threats and vulnerability challenges in FL systems	X	X	√	X	X	√
[87]	Summarized the most used defense strategies in FL	X	X	√	X	X	X
[88]	Developed protocols and platforms to help industries in need of FL build privacy-preserving solutions	X	X	√	√	X	X
[89]	Performed a systematic literature review on FL from the software engineering perspective	√	X	√	X	X	√
[90]	Highlighted FL’s applications in wireless communications	X	X	√	X	X	√
[50]	Discussed the basics and problems of FL for edge networks	X	√	√	√	X	√
[91]	Analysis of FL from the 6G communications perspective	X	X	√	X	X	√
[92]	Survey on FL-powered IoT applications to run on IoT networks	X	√	√	X	X	√
[93]	Survey on the FL-enabled IIoT	X	√	√	√	X	√
[94]	Survey on FL implementation in wireless communications	X	X	√	X	X	√
[95]	Analysis of FL’s potential for enabling a wide range of IoT services	X	√	√	√	X	√
[96]	FDL application for UAV-enabled wireless networks	X	√	√	√	X	√
[97]	Survey on the implementation of FL and challenges	X	√	√	X	X	√
[98]	Review of FL for vehicular IoT	X	√	√	√	X	√
[99]	Survey on FL for the future of digital health	X	√	√	√	X	√
[100]	Survey on applications of FL to autonomous robots	X	√	√	√	X	√
[101]	Review of FL technologies within the biomedical space and the challenges	X	√	√	√	X	√
[102]	A tutorial on FL in the domain of communication and networking	X	√	√	X	√	√
[65]	Analysis and design of heterogeneous federated learning	X	X	√	X	X	√
Our Paper	A systematic review on FL implementation in the EC paradigm	√	√	√	√	√	√

Note: √ = does include; CS: Case Studies; X = does not include; HWR: Hardware Requirements; SLR: Systematic Literature Review; FRD: Future Research Directions; EC: Edge Computing; FL: Federated Learning.

## 4. Research Methodology

The SLR is a well-known method for evaluating papers that come from reliable sources. As the name implies, the SLR aims at determining, interpreting, and evaluating research answers that correspond to the defined research questions. Therefore, the aim of this paper is to review the most recent works of literature by using the SLR research methodology. To retrieve the primary study results, we performed both manual and automatic searches. We analyzed the implementation of federated learning in an edge computing environment based on its impacts, support, applications, and challenges. The primary studies were subjected to quality assessment to capture the best results in the study. We applied both backward and forward snowball approaches to find the most relevant results. To reduce bias in research, researchers must adhere to a predefined protocol. A thorough review of all studies included in an SLR identifies existing research gaps and provides a basis for conducting further investigations and further elucidating the new phenomenon. We utilized the research methodology depicted in Figure 4 [103] to build the SLR, which consisted of five stages. The first step was to formulate the research questions, and the second was to select sources and search strategies. In the third step, results were selected for inclusion and excluded from consideration. The analysis and synthesis of the research papers were discussed in the fourth and fifth steps of the research methodology adopted in this paper, respectively. Furthermore, various software tools, including Excel, were used for the analysis phase.

### 4.1. Research Question Formulation (Stage I)

Several Research Questions (RQs) might arise during the implementation of FL in the EC scheme. The main purpose of this section is to describe the potential RQs and the methodology we followed to solve the problems. Therefore, answering the following RQs is the purpose of this review paper.

RQ-1: What are the architecture and components of FL? The goal of this question is to clarify the structures and elements of the FL design. This question helps readers understand the architecture types for designing FL in the EC infrastructure.

RQ-2: What are the hardware requirements and existing frameworks for FL implementation in the EC environment? The main objective of this question is to explore and discuss the hardware requirements of computing devices to implement FL algorithms. Moreover, the existing frameworks for FL schemes were reviewed. The goal of this research question helps readers easily understand the current hardware requirements of the state-of-the-art framework to implement an FL scheme in the EC paradigm.

RQ-3: What are the applications of FL in the EC environment? This research question intends to identify the state-of-the-art research regarding the applications/roles of FL in the EC environment.

RQ-4: What are the research challenges of FL implementation in the EC paradigm? This research question aims to underline the challenges of FL implementation in the EC environment.

RQ-5: What are the state-of-the-art research solutions proposed to mitigate the challenges analyzed in RQ-4? This question intends to find the state-of-the-art research trends to mitigate the challenges that arise in the implementation of FL in the EC paradigm.

RQ-6: What are the possible case studies to analyze FL implementation in the EC paradigm? This question intends to find the FL implementation in different areas.

RQ-7: What are the potential open research issues and future directions in FL implementation in the EC context? This question aims to define the open questions and research directions in FL in the EC approach. Consequently, answering this question encourages researchers to understand the current research findings and trends in FL in EC.

### 4.2. Source Selection and Strategy (Stage II)

Our review included papers published between the 1 January 2016 and 10 October 2021 time frame that deployed FL schemes in the EC context. We searched through the following search engines and databases: (i) Scopus, (ii) ScienceDirect, (iii) Springer Link, and (iv) ArXiv, which offered excellent coverage of the topics under study. Our search strategy revolved around the terms “federated learning” and “edge computing” and included synonyms and abbreviations as supplementary terms such as “federated ML, federated artificial intelligence, federated AI, federated intelligence, federated training” and “Edge Network, Edge Node, Edge Device” to increase the search results.

### 4.3. Inclusion and Exclusion Methods (Stage III)

We considered that we included the majority of the literature on EC, FL, and FL implementation in the EC context. We devised search strings for each primary source to review the title, abstract, and keywords. After completing the first draft of the search strings, we examined the results of each search string on each database to check the effectiveness of the search strings. The initial search found 882 papers with 343 from Scopus, 116 from ScienceDirect, 202 from Springer Link, and 221 from ArXiv. In addition, using the advanced search criteria, we limited the source type to the English language, journal, conferences, and preprint literature types. We ended up with 204 papers after screening, excluding, and eliminating duplicates.

### 4.4. Analysis and Synthesis (Stage IV)

At this stage, we classified and processed the existing literature that reported the implementation, architecture, and applications of FL in the EC paradigm. This resulted in six classes, i.e., challenges (including communication efficiency, privacy and security, client selection and scheduling, heterogeneity, and service pricing), case studies, the background and fundamentals, applications, and the architecture and framework, as shown in Figure 5. Furthermore, we provide the total number of articles every year since the birth of FL (i.e., 2016) and depict this in Figure 6. In addition, Figure 7 shows the total number of publications per year in the scope of the subject bases. Furthermore, Figure 8 shows the total number of articles in the subject area of the publication on an annual basis.

### 4.5. Reporting and Utilization of the Findings (Stage V)

As shown in Figure 6, the majority of the papers have been published since 2020. This implies the publication rate and field of interest have grown dramatically. The majority (57.3%) of the literature explores the challenges that edge federated learning can face in an edge computing environment. Researchers need to recognize today’s challenges and solutions to anticipate future research directions and tasks. The most frequently explored challenges were heterogeneity, communication efficiency, security, privacy, and client selection. The second-most surveyed topic specifically in the last couple of years was open-source and licensed frameworks for federated learning architecture implementation on an edge network, with 12.7% of the surveyed papers in the selection. The literature describes the various federated learning architectures, as well as a framework to implement them in an edge computing environment. Furthermore, researchers have analyzed and implemented the collaborative learning paradigm for different use-cases such as healthcare, smart homes, smart transportation, and autonomous systems. Due to this, we observed that approximately 12.7% of the papers published since 2016 in the field of distributed learning models focused on the use-cases of the FL model for edge computing. In addition, 10.4% of the works in the literature describe the background and fundamentals of FL in EC. These works aimed to explore the fundamental principles of edge computing and the learning models that reasonably fit into this field. Moreover, articles discussing FL applications in EC accounted for 6.9% of the publications. Federated learning is discussed in these publications for various edge computing applications, including malware detection, anomaly detection, computation offloading, content caching, task scheduling, and resource allocation.

There have been a number of articles published in the past six years in the fields of edge computing and federated learning, as illustrated in Figure 7. The statistics indicates that since the advent of federated learning in 2016, the number of publications related to federated learning in an edge network grew dramatically in 2020 and 2021. Figure 8 depicts the total number of publications annually since 2016 on the four subject bases: case studies, background and foundations, architecture and frameworks, and challenges. Over the past three years, the majority of papers addressing this topic have focused on implementation challenges and potential solutions of federated learning in an edge computing paradigm. As shown in the figure, federated learning architectures and frameworks have also been a hot topic since 2019 in the area. In addition, there are federated learning case studies based on the edge computing paradigm such as in autonomous systems, smart healthcare, and smart transportation. Moreover, the RQ’s are answered in Section 5, Section 6 and Section 7. Figure 5 shows the taxonomy of federated learning in edge computing to analyze the state-of-the-art challenges and solutions we followed in this survey paper. We categorized the existing literature into twohigh-level classifications based on the objectives of the papers. The background and fundamentals include edge computing, deep learning, and federated learning implementations in edge networks. Architectures and frameworks cover potential architectures for federated learning and frameworks to implement it in an edge computing paradigm. Furthermore, applications deal with the possible applications of federated learning in an edge computing setting. In addition, challenges such as communication efficiency, privacy, security, and service pricing are discussed regarding current challenges and potential solutions in the distributed learning paradigm. Case studies are use-cases to demonstrate the application of federated learning implementation in an edge computing paradigm.

## 5. Review and Analysis of Federated Learning Implementation in Edge Computing

This section first describes the FL architecture in the EC paradigm. Following that, we go over the FL protocols and frameworks. In addition, we describe the hardware requirements for FL edge computing devices. Finally, we discuss the challenges and cutting-edge solutions for FL implementation in the EC paradigm. Furthermore, for the FL implementation in the EC environment analysis, this section introduces a taxonomy that includes the studies to be discussed in this review. Figure 5 summarizes the possible classification of FL implementation in the EC paradigm. The classification is based on: (i) Background and fundamentals, which have already been covered in Section 2, (ii) FL applications for the edge computing framework, (iii) implementation challenges for FL in the edge computing paradigm, (iv) FL architectures and frameworks for the edge computing environment, as well as (v) case studies of FL implementation in the edge computing paradigm, discussed in Section 6.

### 5.1. Federated Learning in Edge Computing: Protocols, Architectures, and Frameworks

It is convenient to begin with the network protocol to gain an understanding of the system architecture. To improve overall system performance, the authors of [14] proposed an FL protocol at the system level. The communication protocol deals with the overall FL training process. Thus, it considers the status of the communication between the server and devices, such as communication security, unstable device connectivity, availability, and communication security. The FL server, which is a cloud-based distributed service, and end devices, such as phones, participate in the protocol. When the device tells the server that it is ready to perform an FL operation for a given FL population, the server performs the operation. Learning problems or applications are identified by a globally unique name in an FL population. Tasks associated with FL populations include training with the provided hyperparameters or testing the trained models against local data. In a given time window, the server typically selects a subset of a few hundred devices from the potentially tens of thousands of devices available to it. This subset is used to process a specific FL task. This interaction between devices and a server is referred to as a round. Devices remain connected to the server throughout the round. The server specifies which computations to perform on the selected devices. To do this, it uses an FL plan that contains a TensorFlow graph and instructions on how to execute it. Once the round is set up, the server sends an FL checkpoint with the current global model parameters and any other required information to each participant. Each participant sends an FL checkpoint to the server based on its local dataset and global state. The FL parameter server updates its global state, and the process repeats. Figure 9 illustrates the communication protocol used to develop a global singleton population model in each training round, which consists of three phases:Selection: Devices that meet the eligibility criteria check in to the server on a regular basis through bidirectional communication streams. The availability of clients is kept track of through the stream: whether or not anything is alive and to organize multi-step communication. Furthermore, an FL parameter server selects a subset of active clients for participation in a training round, and they perform a specified FL task based on a defined client selection algorithm such as FedCS [20]. Subsection C discusses the client selection methods and challenges;Configuration: For the selected devices, the FL server configuration varies depending on the aggregation method selected, such as simple [15] or secure [104]. An FL plan and an FL checkpoint are sent to each selected device along with the global model;Reporting: The parameter FL server waits for updates from the participating clients. As updates arrive, the server aggregates them using predefined algorithms such as FedAvg [15] and instructs the reporting devices when to reconnect. When a sufficient number of clients are connected over time, the federated training is accomplished under the control of the server, and the server’s global model will be updated; otherwise, the round will be abandoned. The model update is frequently sent to the server via encrypted communication. To eliminate objective inconsistency, the authors of [105] proposed FedNova as a normalized averaging method.

A couple of papers such as [14,106] described the FL architecture and protocols. For example, Kiddon et al. [14] explained the architecture both on the device participating in FL and the design of the FL server. According to the proposal, we can describe the overall architecture by categorizing it into two main segments: the participating device and FL server architectures.

#### 5.1.1. Architecture of Participating Device

This architecture defines the software design that runs on an FL device, which consists of three core components: application process, example store, and FL runtime. The first responsibility of a participating device in on-device learning is to keep a repository of locally collected data for model training and evaluation. By implementing an API, applications are responsible for making their data available to the FL runtime as an example store. An example store for an application could, for example, be a SQLite database that records action suggestions shown to the user and whether or not those suggestions were accepted. The FL runtime will access an appropriate example store to compute model updates when a task arrives at the device.

#### 5.1.2. Architecture of the Federated Learning Server

The FL server is designed to handle many different scales of population and one or more dimension. Rounds can have thousands of participants or hundreds of millions, so the server must be able to handle them. From a few dozen clients to hundreds of millions, the server must be able to handle FL populations of all sizes. Each round can also contain updates as large as ten megabytes, and the size of the updates can vary from kilobytes to megabytes. Depending on when devices are idle and loading, the amount of traffic flowing through a given geographic region can vary greatly throughout the day.

The actor programming model is at the heart of the FL server. Among the principal actors are:Coordinators: Trains are synchronized globally and flow in lockstep by these top-level players (one for each population). Based on the number of tasks scheduled for FL, the coordinator receives information about the number of devices connected to each selector;Selectors: This accepts and forwards device connections. When the coordinator starts the master aggregator and a set of aggregators, the selectors are instructed to send a subset of their connected devices to the aggregators. This technique allows the coordinator to efficiently assign devices to FL tasks regardless of the number of available devices;Master aggregators are spawned to complete the work according to the number of devices and the update size. To balance the number of devices and the update size, they scale as needed.

Furthermore, the authors of [106] described the software architecture design concerns in building FL systems. The architecture consists of four components: client management patterns, model management patterns, model training patterns, and model aggregation patterns. The client management patterns manage the client devices’ information and their interaction with the central server. In addition, the model management patterns include model transmission, deployment, and governance such as message compression, model versioning, and model deployment. The multitasking model trainer, which handles both model training and data preprocessing, is one of the model training patterns. Model aggregation patterns, on the other hand, are model aggregation design solutions that can be used for a variety of purposes. The authors defined four types of aggregators: asynchronous, decentralized, hierarchical, and secure. An asynchronous aggregator’s goal is to reduce aggregation latency and improve system efficiency, whereas a decentralized aggregator’s goal is to improve system reliability and accountability. The hierarchical aggregator is used to improve model quality and optimize resources. The secure aggregator is designed to protect the models’ security.

The performance of the system is determined not only by the FL architecture, but also by the architecture of the edge network. FL services, on the other hand, rely heavily on complex software libraries. The training model requires not only the computing power of edge devices, but also the ability to respond quickly to services offered by the edge computing architecture. As a result, an appropriate combined EC and FL architecture is required to deal with the coordinated integration of computing, networking, and communication resources. For protection against degradation, it is both necessary to utilize computation virtualization and to integrate network virtualization and management technologies. SDN and NFV are new technologies that aim to improve resource management and orchestration. FL is also a key enabler for implementing a virtualized environment for resource management and orchestration [25,107].

#### 5.1.3. Open-Source Federated Learning Frameworks

FL is actively being developed, and several open-source frameworks are currently being used to implement it. Managing and analyzing a large amount of collected data from edge nodes or devices is one of the challenging issues in the FL-enabled environment. Thus, FL frameworks help engineers to ease the use, develop, and enhance the accuracy and performance of learning models. In this subsection, we discuss the open-source FL frameworks recently developed for FL:TensorFlow Federated (TFF): TFF [108] is an open-source framework for decentralized ML and other computations. TFF was created to enable open research and experimentation with FL by Google. TFF’s building blocks can also be used to implement nonlearning computations such as federated analytics. TFF’s interfaces are divided into two layers: (i) FL API and (ii) Federated Core (FC) API. The FL API offers a set of high-level interfaces that allow developers to apply the included implementations of federated training and evaluation to their existing TensorFlow models. Furthermore, the FC API is a set of lower-level interfaces for expressing novel federated algorithms in a strongly typed functional programming environment by combining TensorFlow with distributed communication operators. This layer also serves as the foundation upon which FL is built;Federated AI Technology Enabler (FATE): The FATE [109] project was started by Webank’s [110] AI Department to provide a secure computing framework to support the federated AI ecosystem. It uses homomorphic encryption and multi-party computation to implement secure computation protocols (MPCs). It supports the FL architecture, as well as the secure computation of various ML algorithms such as logistic regression, tree-based algorithms, deep learning, and transfer learning;Paddle Federated Learning framework (PaddleFL): PaddleFL [111] is a PaddlePaddle-based open-source FL framework. Several FL strategies will be provided in PaddleFL, including multi-task learning [112], transfer learning [113], and active learning [114], with applications in computer vision, natural language processing, recommendations, and so on. PaddleFL developers claim that based on Paddle’s large-scale distributed training and elastic scheduling of training jobs on Kubernetes, PaddlePaddle can be easily deployed on full-stack open-sourced software;PySyft framework: PySyft is an MIT-licensed open-source Python project for secure and private deep learning. Furthermore, it is a PyTorch-based framework for performing encrypted, privacy-preserving DL and the implementation of related techniques such as Secure Multiparty Computation (SMPC) and Data Privacy (DP) in untrusted environments while protecting data. PySyft is designed to retain the native Torch interface, which means that the methods for performing all tensor operations remain unchanged from PyTorch. When a SyftTensor is created, a LocalTensor is created automatically to apply the input command to the native PyTorch tensor. Participants are created as virtual workers to simulate FL. As a simulation of a practical FL setting, data in the form of tensors can be split and distributed to virtual workers. The data owner and storage location are then specified using a PointerTensor. Model updates can also be retrieved from the virtual workers for global aggregation;Federated Learning and Differential Privacy framework (FL& DP): A simple FL and DP framework has been released under the Apache 2.0 license. FL is an open-source framework [115]. Granada’s Andalusian Research Institute for Data Science and Computational Intelligence developed the framework of Sherpa.AI. This framework uses TensorFlow Version 2.2 and the SciKit-Learn library to train linear models and clusters;LEAF: LEAF [116] is an FL benchmarking system that has applications in FL, multi-task learning, meta-learning, and on-device learning. It consists of three parts: (1) a collection of open-source datasets, (2) a set of statistical and system metrics, and (3) a set of reference implementations. Because of LEAF’s modular design, these three components can readily be integrated into a variety of experimental workflows [117]. The “Datasets” module preprocesses and transforms the data into a common format that can be used in any ML pipeline. The “Reference Implementations” module in LEAF is a growing repository of common federated techniques, with each implementation generating a log of various statistical and system characteristics. Through LEAF’s “Metrics” module, any log created in a proper format can be used to aggregate and analyze these metrics in a variety of ways.

#### 5.1.4. Proprietary Federated Learning Frameworks

FL technologies are applied in a variety of ways, including open-source frameworks. Some major industry leaders have created proprietary libraries that are not open-source and are only available under a limited license. In addition to the open-source FL frameworks, there are proprietary frameworks from leading IT companies. NVIDIA, for example, added FL support to its NVIDIA Clara Train SDK. IBM is working on a framework called IBM Federated Learning [118]:IBM Federated Learning Framework: A Python framework for FL in an enterprise environment is IBM Federated Learning. IBM distributes the framework under a license that restricts its use [119]. One of the most salient features of the framework is the large number of ML algorithms it contains. Besides NN, linear classification, and decision trees (ID3 algorithm), it also supports K-means, naive Bayes, and reinforcement learning algorithms. IBM FL integrates libraries such as Keras, PyTorch, TensorFlow, SciKit-learn, and RLlib;NVIDIA Federated Learning Framework [120]: It is not possible to open source the entire framework of the NVIDIA Clara Train SDK because of the restrictive license under which it has been released. For NVIDIA Clara Train SDK, FL requires CUDA 6.0 or later. It supports TensorFlow [121], TResNet [122], and AutoML, making the development of models easy and intuitive. To track the progress of model training, the software uses a centralized workflow between the server and the clients. The original model is then sent to each client.

### 5.2. Hardware Requirements for Implementing Federated Learning in an Edge Computing Environment

DL is capable of a wide range of advanced tasks, including image classification/object detection, audio/speech recognition, and anomaly detection, which require huge computational resources. Edge devices, on the other hand, have limited computational resources, memory footprints, and power consumption. As a result, before implementing FL in the EC paradigm, it is necessary to research the most recent EC hardware requirements and gaps to fill. We were inspired by this idea to conduct a brief investigation into the EC hardware requirements, which we present in Table 3.

The advancement of DNN collaborative training on an edge network such as FL encourages industries to accelerate hardware to support the computational workloads and storage required. These include growth in Central Processing Units (CPUs), Graphics Processing Units (GPUs), and new Application-Specific Integrated Circuits (ASICs) designed to support DNN model computation. With a high depth of an NN model, the DNN structure becomes more complex. To train and test the model, billions of operations and millions of parameters, as well as significant computing resources are required. A requirement of this nature poses a computational challenge for General-Purpose Processors (GPPs). As a result, hardware accelerators to improve the performance of the DNN model can be explored by categorizing them into three edge hardware types for FL:GPU-based accelerators: A GPU is an efficient computing tool because it is able to perform highly efficient matrix-based operations coupled with a variety of hardware choices [123]. The GPU accelerator, on the other hand, uses much power, making it difficult to use on cloud servers or battery-powered devices. In addition to its GPU-based architecture, the NVIDIA Tensor Core architecture runs large amounts of computations in parallel to increase throughput and efficiency. NVIDIA DGX1 and DGX2 [124] are two popular GPU-based accelerators that offer accelerated deep learning performance. There are also Intel Nervana Neural Network Processors [122], two different GPU accelerators that can be used for deep learning training (NNP-L 1000) and the deep learning inference (NNP-I 1000). Mobile phones, wearable devices, and surveillance cameras enable rapid deployment of DL applications, making them even more valuable near the venue. There are ways to perform DL computation on edge devices without moving them to the cloud, but they have limited computing power, storage, and power consumption. Several academics are developing a GPU accelerator for edge computing to solve the bottlenecks. Reference [125], for example, described ARM Cortex-M microcontrollers and developed CMSIS-NN, a set of efficient NN kernels. CMSIS-NN reduces the memory footprint of NNs on ARM Cortex-M processor cores, allowing the DL model to be implemented in IoT devices while maintaining standard performance and energy efficiency;FPGAs-based accelerator: Although GPU solutions are frequently used in cloud computing for DL modeling, training, and inference, similar solutions may not be available at the edge due to power and cost constraints. Furthermore, edge nodes should be able to handle numerous DL compute requests at once, making the use of lightweight CPUs and GPUs impracticable. As a result, edge hardware based on FPGAs is being investigated for edge DL. As a tradeoff for low speed, FPGAs are more energy efficient than GPUs when computing machine-learning algorithms. FPGAs are increasingly vying with GPUs for the implementation of AI solutions as the market evolves. It is estimated that FPGAs are 10-times more power efficient than GPUs, according to Microsoft Research’s Catapult Project [126]. In terms of FPGA-based accelerators, Microsoft’s Project Brainwave is outstanding. It is a high-performing distributed system with soft DNN engines rendered with Intel’s Stratix FPGAs that use real-time, low-latency artificial intelligence [127]. However, it requires a significant amount of storage, external memory and bandwidth, and computational resources on the order of billions of operations per second. Therefore, as FL demands a sufficient storage size for local datasets, further research is required for FL implementation requirements in FPGA-enabled edge devices;ASIC-based accelerator: FPGAs have higher DL inefficiency and require more complex programmable logic, while ASIC architectures for DL have higher power efficiency, but lower reconfigurability. Either way, ASICs are more suitable for DL applications due to their high DL efficiency and programmability. ASICs still provide a much higher overall efficiency than FPGAs for simple algorithms even though FPGAs can reduce the power consumption in computing by optimizing ML algorithms to the hardware design. ASICs introduce complex logic, while FPGAs introduce programmability, increasing the hardware design costs. Moreover, FPGAs have a limited frequency of 300 MHz, four- to five-times less than typical ASICs [123]. Furthermore, due to their reduced network overhead and off-chip memory access characteristics, ASICs are increasingly being used in EC by academia and industry. They also support DL training with a low power and processing time [128]. EdgeTPU [129] is a Google-developed open, end-to-end infrastructure for implementing AI solutions that is a prime example of an ASIC-based accelerator installed in an EC. ASIC-based accelerators are hence promising hardware components for implementing FL in EC devices. Although it has already been tried for several DL applications in EC, such as object identification and recognition [130] and emotion recognition [131], more research is needed to understand the impact of ASIC-based accelerators for FL implementation in EC. However, ASIC-based accelerators have a long development cycle and are not flexible to cope with varying DL network designs.

### 5.3. Federated Learning Applications in an Edge Computing Environment

FL was described in the preceding sections as an EC-enabling technology that facilitates collaborative learning over networks. Additionally, FL allows for decentralized decision-making and the customization of task assignments to each network node, providing multiple applications to the EC field. In the following, we study the main approaches that employ FL application to solve challenges in the EC area. Despite the fact that various studies demonstrated that FL applications including sensitive time such as cancer detection [73] and COVID-19 detection [74,75] applications are drawing attention, we focus on three primary FL applications in EC, which are outlined below.

#### 5.3.1. Computation Offloading and Content Caching

We can bridge the gap between cloud capacity and edge device requirements by bringing edge intelligence to the edge. This improves the QoS and enables the delivery of content. Furthermore, EC leverages the computational resources everywhere in the network using relevant computation offloading schemes, therefore intelligent decision-making to cache or not to cache, when and how task computation offloading is required to enhance the efficiency of communication and computation resources as these have a direct impact on the QoS and QoE of the system performance. Researchers and engineers have been committed to intelligent content cache delivery and computation offloading schemes in the EC environment by leveraging DL for the last decade [76,77,79,132,133]. However, the traditional centralized cognitive system would not be feasible for the following reasons:In the case of EC implementations that involve massive clients, Big Data training is uploaded to a central cloud server for model training through an uplink channel (i.e., wireless link). This introduces additional burdens and congestion on the uplink channel;The uploaded training dataset to the server may be sensitive to privacy, such as patient history data, which results in potential privacy violations;Consider that we need to train the DL model at the end device such as mobile phones, tablets, wearable things, and implanted sensors to prevent privacy. However, the model training demands intensive computational capacity and energy to find the optimal solution as in the SGD convergence algorithm. It uses a large amount of data factors and parameters over a large scale. Thus, DL training at the resource-constrained edge introduces extra energy consumption and long processing times;The conventional centralized training of a DL model fails to handle non-IID and unbalanced data distributions. However, the data distribution in the EC environment depends on several conditions such as the location and amount of data. In addition, the performance of DL usually deteriorates with weak consideration of both data and network state heterogeneity.

FL has been used to deliver intelligent computation offloading and content cache decisions since its introduction to address these challenges. It has become increasingly feasible to cache and deliver mobile content using middle servers (or middleboxes, gateways, or routers). Users can thus easily use the same content without duplicating transmissions by eliminating redundant data from remote cloud servers. A library of content files placed in the edge network from all clients may be requested in the system. Based on the content popularity (i.e., the probability distribution of contents requested from all clients) [134], there is a common interest of all users in the network. Edge node agents determine whether local content shall be replaced based on whether they should cache or not.

The authors of [135] presented a dynamic cache allocation scheme FedCache by leveraging FL in edge networks. In [136], a system that combines IoT devices, edge nodes, the remote cloud, and blockchain was investigated for content caching. The authors also suggested a blockchain-assisted FL compression method, the CREAT algorithm, to predict content files. Furthermore, the combination of FL and blockchain was used to improve data security and shorten the time it takes to upload data. Furthermore, References [137,138] also proposed a proactive caching scheme by leveraging FL in the edge computing paradigm. Edge devices typically have limited computational power and are resource constrained. Therefore, they can use the uplink channel to offload their computation tasks or complete them locally. Therefore, intelligent decision-making is required to perform the offloading process locally [134]. Ren et al. [139] demonstrated FL-enabled computation offloading by deploying multiple DRL agents in distributed IoT devices to indicate their decisions. FL empowers distributed DRL training and reduces the transmission cost between the IoT devices and edge nodes. Furthermore, S. Shen et al. [140] proposed DRL and FL-enabled computation offloading algorithms in the IoT-based edge computing environment. The authors of [141] considered the context information of the application, requests, sensors, resources, and network, which had a significant impact on the offloading decisions. Moreover, they leveraged FL and DRL for context-aware offloading in MEC with multiple user. Compared to the previous works, the context-aware offloading algorithm is better in terms of energy consumption, execution cost, network usage, delay, and fairness.

Coupling service caching and computation tasks in a resource-constrained (i.e., limited storage and capacity) edge device is among the challenging issues in the field. S. Zhong et al. [141] proposed the GenCOSCO algorithm for cooperative service caching and computation offloading, which improved the QoS of the system while reducing the average time consumption of task execution. In addition, the authors took into account the heterogeneity of task requests, application data pre-storage, and base station cooperation when developing the GenCOSCO algorithm. Additionally, to optimize caching and computation offloading decisions in a MEC system by taking dynamic and time-varying conditions into account, the authors of [134] proposed using DRL and FL together. Several base stations were used to cover the collection of clients in the MEC system. Cached files and local files were replaced by the DRL agent when caching was enabled.

#### 5.3.2. Malware and Anomaly Detection

Malware is a global threat that has grown in number and diversity with the rapid growth of IoT applications, making threat detection and analysis a critical challenge to address. A major challenge in developing effective malware detection is the increasing diversity of the malware syntax and behavior. As a result, the most important steps in mitigating and preventing severe consequences in the edge computing environment are acquiring knowledge of different threats and developing efficient malware attacks (or cyber-attacks) and anomaly detection schemes [142]. Currently, there are some approaches proposed for malware and anomaly detection and prevention in a different environment. The authors of [143] proposed deep learning-assisted cyber intrusion attack detection using the Gaussian naive Bayes classifier. Furthermore, they used the NSL-KDD [144] dataset and the UNSW-NB15 [145] dataset for model training, which had a shorter detection time and higher classifier accuracy. In [146], a CNN was used to extract an accurate feature representation of data and then classify them using the LSTM model for cyber-attack detection, which can outperform other DL methods for detecting intrusions. DL-based solutions depend on the available datasets to achieve the required detection accuracy. However, the datasets usually contain sensitive or confidential data that would significantly affect environmental or personal security and privacy. In order to circumvent these problems, FL-based attack detection models have been introduced for EC-enabled networks.

The authors of [147] proposed an FL-enabled anomaly detection framework for EC-enabled networks, as illustrated in Figure 10. Each participating client encompasses different IoT devices that produce a set of private training data for anomaly detection. This model contains four building blocks: local training, cloud aggregation, anomaly detection, and global model. All participants perform local model training using their own local datasets. To improve the recognition accuracy, after training the global model, participants send their trained local models to the FL server. Each participant receives an update of the global model based on the data collected by the server. The communication rounds are repeated until the desired performance level is achieved. After the global model is updated, the anomaly detection engine can use it.

Figure 10 illustrates how edge computing applications can detect anomalies or attacks using federated learning. The nodes build local models based on the individual datasets. The updated local models are also sent to the FL server from each selected node, as illustrated by the blue dotted lines in the figure. As a result, global model aggregation occurs in the parameter server (i.e., cloud aggregation). The anomaly detection block includes training and testing the model to develop a global cyber-attack detection model. In the proposed system, the anomaly detection block consists of four components: a pattern learner, a pattern recognizer, a threshold determinator, and an anomaly classifier. When the training has been completed and the model has been configured for testing, the pattern learner receives the global model. In order to determine a suitable threshold value for a specific problem, the pattern recognizer tests the model and evaluates its performance based on the test data. An anomaly classifier generates an inference based on the threshold value. An anomalous data point is detected in the anomaly detection system if it does not pass the threshold set by the determinator. Once this round of training is completed, the updated global model can be deployed on sensors. All IoT devices involved in the first round of training receive the updated global model after the model weights are updated by the anomaly detection engine. This engine is located in the IoT system. A new round of training begins when the weights in the global model are adjusted and the weights in the local model are updated.

Furthermore, the authors of [147,148] leveraged the multi-task learning paradigm to solve multiple tasks simultaneously, while anomaly detection in FL settings is based on commonalities and differences in tasks. Furthermore, the authors of [149] proposed a distributed deep-learning scheme of cyber-attack detection in fog-to-things computing, which they considered in terms of accuracy, detection rate, and scalability, as the metrics of the model. Anomaly detection is not the scope of this survey, and we recommend readers read the papers [150,151,152,153,154], where FL was used an an enabling technology for malware and anomaly detection.

#### 5.3.3. Task Scheduling and Resource Allocation

Edge computing is supposed to support various resource-hungry IoT applications and services with low latency in the edge network, which minimizes the response time and burden of the backhaul link between the edge device and the server. Computationally intensive IoT tasks are sent to neighboring VMs at the edge server to achieve low-latency services. However, edge nodes are generally resource-constrained such as in terms of computation, storage, and network. Thus, task scheduling is an important technique to maintain a Key Performance Indicator (KPI) of the network effect and efficiency to improve the QoS and QoE. However, task scheduling in the edge computing paradigm is challenging for the following reasons. The transmission latency is stochastic due to the dynamic behavior of the link between the end devices and the edge node, making scheduling complex and challenging. Furthermore, in terms of availability, speed, ready time, and response time, edge resources are dynamic and variable, posing design issues for scheduling algorithms. In addition, the task arrival rate, task size, and delay requirements are diverse for various IoT applications, making task scheduling in edge computing more challenging. In order to implement task scheduling in EC, two challenges must be addressed: time scheduling and resource allocation. Time scheduling determines the task execution order, and resource allocation is responsible for assigning tasks to suitable VMs for execution [155].

The authors of [155] leveraged DRL to solve both time scheduling and resource allocation considering the diversity of tasks and the heterogeneity of available resources, and a Fully Connected Neural network (FCN) applied to extract the features. Several studies including [156,157,158,159] showed that DRL has been the popular centralized DL algorithm implemented in EC for the task scheduling and resource allocation scheme. However, due to privacy concerns, edge device owners are hesitant to share their identifiers. For example, sharing their individual Queue State Information (QSI) causes severe privacy consequences. As a result, FL-based task scheduling and resource allocations address this issue.

Wang et al. [160] investigated an optimization scheme to reduce energy and time consumption for task computation and transmission in MEC-enabled High-Altitude Balloon (HAB) cellular networks. HABs perform computation tasks for users with limited capacity and energy. Because the data sizes of the computation tasks vary, the user association policy should be optimized to meet the requirement while consuming the least amount of energy. To overcome these constraints, the authors used an SVM-based federated learning algorithm to map the relationship among the user association, service sequence, and task partitioning schemes. The authors of [161] also suggested a distributed, FL-based combined transmit power and resource allocation scheme for ultra-reliable and low-latency vehicular communication.

### 5.4. Implementing Federated Learning in an Edge Computing Environment: Challenges and Solutions

As explained in the earlier sections, FL shows its increasingly significant role in supporting edge computing services and applications. Despite its great potential, in this section, we would like to discuss the relevant research challenges considered for future FL implementation in edge networks. Here, we focus on a few relevant challenges and their corresponding state-of-the-art solutions in FL, including communication efficiency, managing heterogeneity, security and privacy preservation, and service pricing requirements in the edge network.

#### 5.4.1. Communication and Computation Efficiency

One of the primary goals of FL is to reduce the burden of the communication link in the core network by training a high-quality centralized model with distributed training data across a large number of clients, each with unreliable and relatively slow network connections [16]. In addition to the unreliable wireless and asymmetric channel characteristics, intensive training and several updates are performed between clients and the server to improve model accuracy. Furthermore, complex DL models include millions of parameters in different applications such as image recognition and detection [162,163]. Thus, the updates with large dimensions require a high communication cost and are a bottleneck for training. Therefore, communication efficiency approaches seek to reduce the communication overhead that arises from the exchange of messages between the server and clients that run the model training in a distributed fashion. The key to reducing communication costs in FL in an edge network is to decrease the number of devices involved in the communication, total communication rounds, and model update size. Model updates messages are compressed between the server and the client to minimize the size, referred to as model compression. For FL implementation, compression techniques, specifically the quantization and sparsification compression methods, are used to reduce the update size.

Furthermore, the optimization methods for FL are largely derived from traditional distributed ML optimization [164]. To determine the primary and dual variables, the traditional distributed approach sends the local stochastic gradients to a central node, which then aggregates the gradients from each client. The main bottleneck slowing down the distributed optimization is due to the heavy communication outages caused by this approach. Utilizing local SGD with periodic averaging is one way to reduce the complexity of computations under a budget [16,165]. With local SGD, the model is updated by the client several times via SGD, and an average of the models of the various clients happens at regular intervals. By reducing the synchronization frequency between the working nodes, local SGD can reduce the overhead of distributed learning [166]. Furthermore, the authors of [167] proposed a method that supports multiple local updates on the workers for reducing the communication frequency between workers.

Alistarh et al. [168] proposed Quantized SGD (QSGD), a family of compression schemes with convergence guarantees to improve performance. Similarly, the authors of [169] proposed an FL framework to improve the FL convergence time and training loss, which includes three components: (1) probabilistic device selection for limiting the number of participating devices, (2) a universal FL parameter compression method for reducing the volume of data conveyed at each FL iteration, and (3) a resource allocation scheme for optimizing the usage of the wireless channel. FedPAQ [57] relies on three key features: (1) periodic averaging, where models are updated locally at devices and only periodically averaged at the server; (2) training rounds consisting of only a fraction of the devices; (3) quantized message passing where the edge nodes quantize their updates before uploading to the parameter server. Gradient quantization methods should be used with caution when updating federated models. When training with non-IID data, that is highly distributed, the model divergence increases. A quantized model update leads to many quantization divergences as the training size increases. To compensate for errors in model fitting, one can use the quantization described in [170]. SGDs with reduced variance and quantized SGDs with reduced variance [171] are also essential.

In [172], to reduce the communication cost, the authors proposed a convex optimization formulation to minimize the coding length of stochastic gradients. The key idea is to randomly drop out the coordinates of the stochastic gradient vectors and amplify the remaining coordinates appropriately to ensure the sparsified gradient [163] is unbiased. To solve the optimal sparsification efficiently, a simple and fast algorithm was proposed for an approximate solution, with a theoretical guarantee of sparseness. The authors of [173] introduced a novel Time-Correlated Sparsification (TCS) scheme, which builds upon the notion that a sparse communication framework can be considered as identifying the most significant elements of the underlying model. Dynamic device scheduling has also a great impact on optimizing both the communication and computation costs [174]. Generally, FL implementation in the EC paradigm imposes not only communication cost, but also computing cost. Although local SGD is more communication efficient than distributed SGD, it consumes more power and energy in the participating devices. Therefore, optimum methods are required to study to explore communication and computation efficiency.

#### 5.4.2. Heterogeneity Management

Unlike the traditional distributed optimization, the participating devices and networks are considered in terms of hardware such as CPUs, GPUs, memory, network configuration, and power supply. The diversity in the devices’ hardware and network configurations is known as system heterogeneity. System heterogeneity significantly affects the model aggregation efficiency and accuracy and may lead to the divergence of the optimization. Therefore, several kinds of research have been conducted to mitigate the impact of system heterogeneity in the FL training process. FedProx, a variation of FedAvg, was proposed in [175] to address system heterogeneity by which the device characteristics vary depending on where the federated training is being run. In FedProx, tasks are associated with the resources available on each device based on the work that needs to be performed. Furthermore, the authors of [176] presented a federated learning framework that allows one to handle heterogeneous client devices that couples a parameterized superquantile-based objective. Yang et al. [177] proposed the first empirical study to characterize the effects of heterogeneity in FL using large-scale data from 136k smartphones that could faithfully reflect heterogeneity in real-world settings. Furthermore, they created a heterogeneity-aware FL platform that adheres to the standard FL protocol while taking heterogeneity into account. Similarly, in [178], the extent of device heterogeneity, which is a major contributor to training time in FL, was investigated. Furthermore, they proposed AQFL, an approach that uses adaptive model quantization to homogenize client computing resources.

Traditional distributed learning systems provide access to the entire training dataset to a central server. To obtain an efficient model, the server splits the dataset into subsets and distributes them to participating devices based on the distributions. Due to the fact that the local dataset is only accessible by the data owner, this method is not practical for FL. Another issue that can arise in a federated system is that the amount of participating devices’ local datasets varies. Some have a regular size and a normal distribution, while others are small and have a limited number of data points. Therefore, the clients can frequently generate and collect imbalanced datasets that limit the accuracy of the FL model. Several studies have shown that imbalanced distributed training data lead to a degradation of the accuracy of FL applications. The authors of [179] highlighted the effects of the imbalanced distribution and developed Astraea’s self-balancing FL framework, which mitigates the imbalances. Moreover, the proposed framework relieves the global imbalance by adaptive data augmentation and downsampling, and by averaging the local imbalance, it creates a mediator to reschedule the training of clients based on the Kullback–Leibler Divergence (KLD) of their data distribution. Moreover, the authors of [180] designed new methods for detecting data imbalances in FL and mitigating their effects. In addition, they proposed a monitoring scheme that can infer the composition fraction of the training data for each FL round and designed a new loss function to mitigate the effects of the imbalanced data. Dipankar et al. [181] introduced a new loss function called fed-focal loss to alleviate the effects of data imbalance by redesigning the cross-entropy loss to weight loss designated to well-classified models along the lines of focal loss.

In contrast to traditional DL training, participating devices across all edge networks generate and collect datasets in a non-Identical and Independent Distribution (non-IID) in FL model training, which has the potential to bias optimization procedures in DL model development [182]. The FL statistical model requires two levels of sampling: to access a data point, at the first level a client i∼Q, the distribution over available clients, and extracting examples $\left(x,y\right)∼P_{i}\left(x,y\right)$ from local clients training dataset distribution, where *x* is the features and *y* is the label. Non-IID data in FL mean the difference between $P_{i}$ and $P_{j}$ for different clients *i* and *j*, respectively. This indicates the data on each node being generated by a distinct distribution $x_{t}∼P_{t}$. Moreover, the number of data points on each node may also vary significantly [183]. The non-IID dataset is then one of the statistical challenges of FL, lowering the model accuracy or making the weight parameter drastically different. For neural networks trained on highly biased non-IID data, the FL model accuracy drops by around 55% when each client device trains on only a single class of data [183]. Overcoming problems with non-IID training data is one of the current research topics, and several researchers have worked to mitigate this problem.

To approach the non-IID challenge, Smith et al. [112] proposed a multi-task learning framework and developed MOCHA to address the system challenges in MTL. However, this approach differs significantly from previous work on federated learning. McMahan et al. [16] showed that FedAvg can work with certain non-IID data and the accuracy is drastically reduced. The authors of [184] proposed a strategy to improve training on non-IID data by creating a small subset of data that is shared globally by all edge devices. Similarly, the authors of [180,184,185,186,187] used heuristic approaches that aimed to deal with statistical heterogeneity by sharing local device data or server-side proxy data. However, these methods may be unrealistic, burden network bandwidth, and violate the key privacy assumption of FL.

Furthermore, Li et al. [175] proposed the FedProx framework to address heterogeneity by allowing partial information aggregation and adding a proximal term to FedAvg [16]. In addition, Wang et al. [188] proposed a FedMA framework, an aggregation strategy for non-IID data partition that shares a global model layer in a layerwise manner. In addition, Reisizadeh et al. [189] proposed FedRobust, which assumes that the data follow an affine distribution shift and address this problem by learning the affine transformation. This complicates the generalization if we cannot estimate the explicit affine transformation. Andreux et al. [190] proposed a SiloBN framework that empirically showed that local clients that retain some untrainable Batch Normalization parameters (BN) can improve robustness to data heterogeneity, but did not provide a theoretical analysis of the approach. Xiaoxiao Li et al. [191] proposed an effective method, FedBN, that uses local BN to mitigate feature shift before model averaging, and instead, it keeps all BN parameters strictly local, which is the state-of-the-art solution to mitigate the statistical heterogeneity challenge in FL.

#### 5.4.3. Privacy and Security Preservation

The original intent of FL was to resolve the issues related to data privacy, ownership, and legalization [80]. The fact that models, parameters, and global models are shared with each client presents several risks for exploiting FL’s vulnerabilities in EC environments. Edge FL needs to investigate the existing attacks and solutions to provide a secure and privacy-protected environment for user data. In this section, we describe the security and privacy vulnerabilities in the implementation of FL in the EC paradigm, as well as existing solutions:Security: Since FL has numerous clients for collaborative training and exposure to model parameters, it is vulnerable to various attacks and risks. Therefore, we can analyze the security challenges by identifying the threats/attacks and the corresponding defense/solutions:(a)Attacks: Generally, two main classifications of attacks are identified to manipulate the collaborative learning process in the edge FL: Byzantine and poisoning attacks. Byzantine attack is the attack performed by a fully trusted node that has turned rogue and already has passed all authentication and verification processes, for example when selected participants turn into rogues in the FL learning process. If some nodes are compromised, attacked, or fail, the entire FL system fails [192,193].A poisoning attack is similar to an injection attack in the FL training process. The two types of poisoning attacks are data poisoning and model poisoning. By targeting ML algorithms’ vulnerability and introducing malicious data during training, data poisoning attacks aim to maximize the classification error [194]. Through FL, the client interacts with the server; the client sends the training data and parameters, but the client can also manipulate the training to poison the model. Data poisoning in FL refers to sending false model parameters using dirty samples to train the global model. Malicious clients can also inject malicious data into a local model’s processing in a method known as data injection. Because of this, a malicious agent is able to manipulate the local models of multiple clients and influence the global model. Malicious agents use fake data in data poisoning, but they target global models directly in model poisoning. Research has shown that data poisoning attacks are less effective than model poisoning attacks [194]. When multiple clients use a large-scale FL product, there is a greater likelihood of model poisoning happening. It is generally possible to poison the global model by modifying the updated model and then sending it to the central server for aggregation;(b)Defense: A client may intentionally or unintentionally deviate from the prescribed course of FL training, resulting in abnormal behaviors. Timely detection and aversion of these abnormal client’s behavior is therefore critical to minimize their negative impact [195]. Many solutions have been proposed, mainly related to the safe FL training process. The authors of [196] proposed a detection method for Byzantine attackers using a pre-trained anomaly detection model, a pre-trained autoencoder model running at the server level to detect anomalous model weight updates and identify their originators. Based on their results, they found that detection-based approaches outperform conventional defense-based methods significantly. However, Byzantine-tolerant aggregation methods are inefficient due to the non-identically and independently distributed training data. Moreover, Wei Wan et al. [197] proposed a density-based detection technique to protect against high-rate attacks by modeling the problem as anomaly detection to effectively detect anomalous updates. Due to this, the global model became less adversarial. However, this approach is computationally intensive even though it improves the accuracy of the global model, which will be affected by the convergence of the aggregated model. In fact, model poisoning attacks with limited training data can be highly successful, as demonstrated by [198]. In order to prevent such attacks, several recommendations were proposed. When a participant shares a new version of the global model, the server evaluates whether the updated version can improve the global model. By default, the server marks the participant as a potential attacker, and after viewing updated models for a few rounds, it can decide whether or not the participant is malicious. The second solution allows participants to compare their updated models. Malicious participants can update models that are too different from those around them. Until it can determine whether or not this participant is malicious, the server will observe updates from this participant. Although impossible to prevent, model poisoning attacks can occur because it is impossible to assess the progress of each and every participant when training with millions of participants. Therefore, more effective solutions are needed.Furthermore, the authors of [199] described the use of a learning-rate-adjustment system against Sybil-based poisoning attacks that uses the similarity of gradient updates to regulate learning rates. A system using FoolsGold can withstand the Sybil data poisoning attack with minimal modifications to the conventional FL process and without using any auxiliary information. In other words, using a variety of participant distributions, poisoning targets, and attack strategies, the authors showed that FoolsGold mitigates such attacks. However, they did not propose any convergence-proof defense methods. Further, the authors of [200] presented a method of detecting and eliminating malicious model updates based on spectral anomaly detection techniques exploiting its low dimensions. Both Byzantine and targeted model poisoning attacks include the detection and removal of spectra anomalies occurring on the server side. The central server can detect and remove malicious updates using a powerful detection model, providing targeted defenses. Convergence and computational complexity, however, will need to be analyzed;Privacy: Although FL improves the privacy of participating clients, there may still be a breach of privacy when data are exchanged among servers. During the training process, orchestrating FL servers may extract personal data [80,201]. Privacy threats in FL generally fall into three categories: attacks by affiliation inference, inadvertent data leakage and reconstruction inference, and GANs-based inference attacks. An affiliation inference attack searches for private data in the training dataset, abusing the global model. In such cases, the information about the training dataset is derived by guessing and training the predictive model to predict the original training data. Furthermore, inadvertent data leakage and reconstruction inference is a method to extract information from the FL server by sending model updates from participating edge devices. Since GAN can effectively learn the distribution of training data, GAN-based attacks aim to reconstruct, for example, human-distinguishable images from the victim’s personal dataset [80].Differential Privacy (DP) and secure aggregation approaches are the two most common methods to mitigate privacy issues in FL [104]. Differential Privacy (DP) is a mathematical concept often used in statistical machine learning to overcome personal information leakage during data collection and processing. Thus, numerous works of literature proposed DP as the main privacy-preserving algorithm in FL implementation. The authors of [202] analyzed FL using the DP approach and designed a client selection method to improve privacy for subscribers. Yao Fu et al. [203] analyzed the practicality of DP in FL by tuning the number of iterations performed to optimize model accuracy. Secure model aggregation is a privacy-preserving algorithm that ensures that multiple clients participate in global model training while protecting their data shared with others. Therefore, secure model aggregation is a critical component of FL model training. It can be implemented in different approaches such as homomorphic Encryption [204], secure multiparty computation [205], and blockchain [206]. Recent research has been conducted on blockchain-enabled FL in edge networks. The authors of [207] proposed a data-protecting blockchain-based FL for IoT devices. As DP algorithms are lightweight compared to secure aggregation algorithms, they can be deployed easily in edge computing environments with limited resources. DP algorithms are generally used for less sensitive queries, making them vulnerable to privacy leaks, especially when adaptive queries are needed in applications or services. Although more computationally and storage intensive, secure aggregation algorithms are more secure than DP-based aggregation.

#### 5.4.4. Client Selection and Resource Allocation

The FL server requires sufficient data from participating clients to develop the model with the required accuracy. However, limited wireless resources restrict the number of participating clients in each round. For this reason, in FL model development, a subset of the participating clients takes part in the raining process in each round. Therefore, the participating clients need to be selected and scheduled to participate in the training process in each round to achieve the required performance of the model. There are a couple of works that address this issue and the approaches proposed in the existing works. The authors of [208] described the convergence analysis of federated optimization for biased client selection strategies and quantified how selection bias affects the convergence speed. They also proposed power-of-choice, a communication- and computation-efficient client selection framework that can flexibly bridge the tradeoff between convergence speed and solution bias. Minxue et al. [209] proposed FedGP, an FL framework built on a correlation-based client selection strategy to increase the convergence rate of FL by modeling the loss correlations between clients with a Gaussian process. Similarly, the authors of [210] proposed a Probabilistic Node Selection framework (FedPNS) to dynamically change the probability for each node to be selected, based on the output of optimal aggregation. FedPNS can preferentially select nodes that drive faster model convergence. Furthermore, Bo Xu et al. [211] proposed online client scheduling for fast FL to minimize the training latency of a wireless FL system for a given training loss through client scheduling, instead of assuming that the prior information about the wireless channel state and the local computing power of the clients are available.

#### 5.4.5. Service Pricing

Service pricing approaches determine how client devices interact with the FL server from an economic viewpoint. As a result, federated learning requires economic models to make the process more appealing to clients. Currently, some research works have been presented on the service pricing models in federated learning. The authors of [212] proposed two gradient-based methods to allocate the profit generated by the joint model to individual data providers in FL. In addition, Jiawen K. et al. [213] developed the contract theory to design an effective incentive mechanism for simulating mobile devices with high-quality (e.g., high-precision) data to participate in federated learning. Moreover, the authors of [214] adopted the relay network to construct a cooperative communication platform to support the transmission and trading of model updates. Similarly, Yutao J. et al. [214] investigated the design of two auction mechanisms for the federated learning platform to maximize the social welfare of the federated learning services market.

## 6. Federated Learning in Edge Computing: Case Studies

Case studies are presented in this section to help better understand the requirements and advantages of FL implementation in the EC setting across different applications. Following are the use cases and solutions that were identified to demonstrate the implementation of FL in EC:

### 6.1. Smart Healthcare

DL-enabled EC was introduced to reduce overburdening on cloud servers and also to enable real-time analysis of smart home and medical data. In [215], the authors described ResiDI, an intelligent decision-making system for a residential distributed automation infrastructure utilizing wireless sensors and actuators. Similarly, leveraging fog computing and artificial intelligence plays a crucial role in smart home applications [216,217,218]. However, the modern DL models have millions of parameters that need to be learned from sufficiently large curated datasets to achieve clinical-level accuracy and decision-making precision for residential data. This data streaming from the medical devices to the server still adds backbone load and latency, which can be fatal for clinical decisions. Moreover, in healthcare, data privacy is one of the most important issues for an organization and government legislation, which cannot be achieved with traditional DL. In FL, training is performed at the individual client level, and then, the local weights of each client are sent to the server. The server collects the updated local weights and computes the new global weights. Then, the client downloads the global weights from the server and continues the training process. Therefore, FL aims to mitigate the above problems by training algorithms collaboratively without sharing the data themselves. By combining federated learning and data aggregation with a strong focus on security and privacy, the authors of [219] proposed IOTFLA, an architecture for smart homes. Furthermore, FL can be used to solve privacy issues and reduce the risk of data breaches for clinical information as it does not require data transmission and centralization. Thus, FL is a fundamental technology for future smart healthcare [99]. As described in Figure 11, patient data are held locally, and the trained model is sent to the parameter FL server for the purpose of aggregation without sharing the local medical data. A nurse or doctor can make medical decisions based on the global model.

The successful implementation of FL in an EC-enabled healthcare infrastructure could therefore hold significant potential for enabling precision medicine at scale, leading to models that provide unbiased decisions, optimally reflect an individual’s physiology, and respond to rare diseases, while addressing governance and privacy concerns. For this reason, several proposals have been made to describe the possible solutions of FL in EC-enabled smart healthcare. The authors of [220] analyzed and implemented differential privacy techniques to protect patient data in a federated learning facility to ensure data security of patient brain tumor segmentation. They studied differential privacy techniques for protecting patient data in federated learning environments. Exploiting the BraTS dataset, they used federated learning systems to model brain tumor segmentation. According to the experimental results, the model performance correlated negatively with privacy protection costs. Similarly, N. Rieke et al. [99] highlighted the challenges and considerations of edge FL-enabled infrastructure as it relates to the evolution of digital health. Privacy, security, performance, level of trust, heterogeneity of data, traceability, and accountability were among the challenges and limitations discussed by the authors. In addition, the data in these settings are sensitive and need to be protected accordingly. The system architecture enables participating health institutions and hospitals to securely and privately train their models using their distributed computational resources. However, it has not been examined whether the security or privacy of clinical data was sufficiently protected when a non-trusted participant acted as an intermediary and was provided access to it via the FL system, potentially violating clinical data privacy and security. In this way, FL can guarantee patient privacy when designing medical AI systems. However, it may be possible to estimate data throughout the model aggregation phase even though the data have not been obtained or exposed in FL. For the system design, to preserve privacy against a variety of modern privacy attack methods, the authors pointed out that other privacy preservation techniques, such as DP, SMC, and homomorphic encryption, may be needed. In addition, the authors of [101] provided an overview of federated learning technologies, particularly in the biomedical domain. Although the health institutions collaborated, all training data remained at the original institutions. The authors explored ways to deal with the statistical problems, system challenges, and privacy concerns that are associated with federated learning, as well as its implications and opportunities for healthcare. Furthermore, through the use of three benchmark medical datasets, the authors of [221] aimed to evaluate the reliability and performance of FL. The authors developed an FL framework with APIs in Django and AWS using the benchmark datasets MNIST, MIMIC-3, and ECG from Physionet. While their FL model performed as well as the more traditional centralized model, it had better privacy.

### 6.2. Unmanned Aerial Vehicles

The use of Unmanned Aerial Vehicles (UAVs) in applications such as surveillance and monitoring, military, medical supply delivery, and telecommunications has increased dramatically over the past decade. By using UAVs as mobile base stations, mobile networks can become more comprehensive, powerful, and energy efficient. In contrast, a UAV can be used for several tasks, including video streaming and delivery. Aside from utilizing vast amounts of data, data-driven DL will improve the efficiency of networks, as well as the quality of service they provide. However, DL-based systems are cloud-based and require UAV data to be sent to and stored on a central server. While transmitting raw data to the processing unit, UAVs consume much energy and network bandwidth. Because of this, it places an enormous burden on the network. In addition, the transmitted raw data may have personal information including the UAV’s location and identity, which may directly affect the UAV’s privacy [96].

Given these challenges, FL emerges as a promising paradigm that aims to protect device privacy by allowing devices to train DL models locally without sending their raw data to a server. Instead of training the DL model on the data server, FL allows devices to perform local training on their own data. As described above, UAVs can be used as Base Stations (BSs) in mobile wireless networks in areas where it is difficult to implement BSs. Mobile devices on the ground can utilize FL to perform distributed DL tasks without relying on a central system. A UAV does not need to receive raw data during training because the devices do not need to send them. Each of the mobile devices uses its own local dataset to train the DL models, which are then sent to an FL-enabled UAV server for further model aggregation. During local model training, the UAV server collects the device-specific parameters, aggregates them, and sends the updated parameters to the associated devices. As a result, raw data remain on the devices, protecting the privacy and reducing network traffic at the same time. Several rounds are performed until a certain level of accuracy is achieved, as shown in Figure 12. Therefore, a UAV network with FL improves the QoS and privacy of clients. For this reason, the topic has attracted many researchers in the field, and some studies have already been proposed [222]. According to the authors of [222], a framework was developed for asynchronous distributed learning on UAV networks. This framework allows for training of models locally without transferring sensitive data to a UAV server. They investigated how to reduce the execution time and accuracy losses in multi-UAV networks that support federated learning. In order to improve the learning efficiency, AFL uses mobile devices with high communication and computation capabilities. However, their approach is still susceptible to data leakage from aggregated gradients. Therefore, cryptographic mechanisms are needed to ensure that FL is aggregated in a secure manner. In order to ensure FL-based UAV network security and privacy, more research is needed. UAV-based exploration was also performed by the authors of [223]. As part of the UAV-aided exploration scenario, they planned to use Ground Fusion Centers (GFCs) to display image classification tasks if multiple UAVs are to be coordinated from a location that is strategic, but inaccessible, such as the top of a mountain, where recharging batteries might not be feasible.

## 7. Open Issues and Future Research Directions

Although several research solutions have been proposed to mitigate the challenges in FL implementation in the EC paradigm, there are challenges that are still unresolved. In addition to the research initiatives reported above, we envision several potential prospective research trends for federated learning implementation in edge computing in the near future, as described below:Multi-model support federated learning training process: In the FL training process, it is assumed that the participating clients update their corresponding model parameters according to a global model. However, clients may wish to train multiple models, even during their idle time. Therefore, decoupling global model aggregation from local training allows clients to use different learning algorithms. For instance, it may be necessary to develop multiple and various models using a federated approach for different purposes. How should the federated learning architecture look? What is the best way to manage this problem? Therefore, appropriate techniques need to be analyzed and implemented;Impact of wireless channel: Edge devices are often connected to edge or cloud servers over unreliable wireless channels. Moreover, we believe that channel characteristics affect the model accuracy. Therefore, studying the impact of the network requirements, especially wireless communication, on the accuracy of federated model training is considered a future research trend. Noise, path loss, shadowing, and fading are all impairments that should be considered in wireless communication systems. In the federated learning process, the communication between clients and parameter servers usually occurs over an impaired wireless channel. This raises some research questions, for example: How does channel fading affect learning? How can it be mitigated?Joint dynamic client selection and adaptive model aggregation: We described the effects of client selection and model aggregation algorithms on resource allocation independently. However, adaptive model aggregation and dynamic client selection for resource allocation considering the non-IID behavior of data, computational power, the data size, network capacity, and link reliability are among the future research works. Therefore, joint dynamic client selection and adaptive model aggregation should be investigated as future research trends. Consider, for instance, the algorithms and their complexities that are suitable for dynamically varying client requests and volatile resources on both the client-side and server-side. In such scenarios, how can we manage the federated learning training process?Adaptive privacy-preserving security solution: We already described the tradeoff between privacy-preserving techniques and communication/computational efficiency. It is critical to develop solutions that protect privacy while supporting a heterogeneity of client devices in terms of hardware and software. Therefore, an adaptive privacy-preserving security solution is a possible research direction;SDN/NFV-enabled federated learning implementation: Software-Defined Networking (SDN) and Network Function Virtualization (NFV) have captured people’s attention and are increasingly redefining the way networks are built and managed for a modern, connected, always-on world. We also believe that SDN/NFV would be a possible research topic;Service pricing in edge federated learning: As we described earlier, service pricing in federated learning in edge computing determines how clients interact economically with the cloud/edge server. However, service pricing in federated learning at the edge has not yet been analyzed and implemented and requires further research;New federated learning approach: The size of the federated learning model is too large to fit on a resource-constrained edge device. Moreover, the training of the federated learning model is too slow to converge and meet the delay requirements in certain delay-sensitive applications. A new federated learning approach is required to achieve the goals dynamically. Therefore, dynamic and adaptive federated learning must be analyzed and implemented for resource-constrained edge devices.

## 8. Conclusions

FL is highly suited for EC applications, as it can take advantage of the processing edge servers’ capabilities and the highly distributed edge devices generating data. FL allows the collaborative framework of a DL model for edge computing network optimization. As a result, it is an enabling technology in edge computing networks. This paper presented a systematic literature review on FL in the EC paradigm with 200 primary studies in this article. In addition, we discussed the challenges in implementing FL at the edge, which includes communication and computation efficiency, heterogeneity, privacy and security, client selection and resource allocation, and service pricing. For the above challenges, we compiled a comprehensive overview of the latest developments and solutions. Furthermore, we described two case studies (i.e., smart healthcare and UAV) relevant to demonstrate the application of FL in the EC environment. In conclusion, we highlighted the open subjects and potential prospective research trends in implementing FL into the EC paradigm, which encourages researchers to extend and develop their existing work.

## Figures and Tables

**Figure 1 sensors-22-00450-f001:**
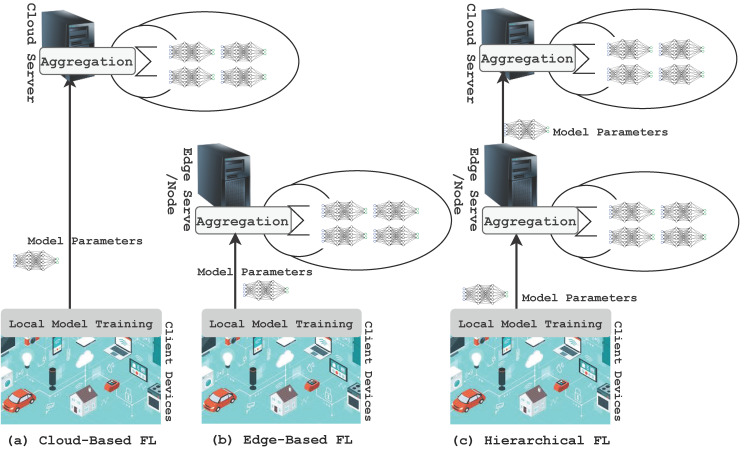
Three federated learning structures: (**a**). Cloud-enabled, (**b**) edge-enabled, and (**c**) hierarchical (client-edge-cloud-enabled).

**Figure 2 sensors-22-00450-f002:**
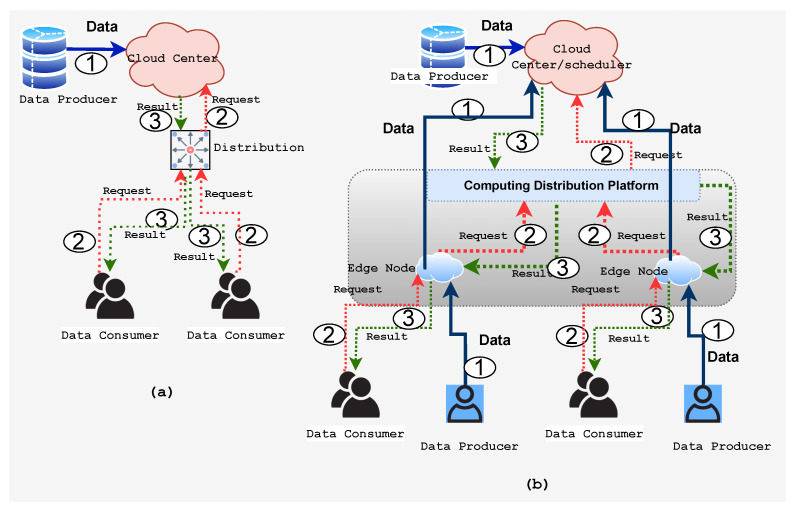
(**a**) Cloud computing paradigm and (**b**) edge computing paradigm.

**Figure 3 sensors-22-00450-f003:**
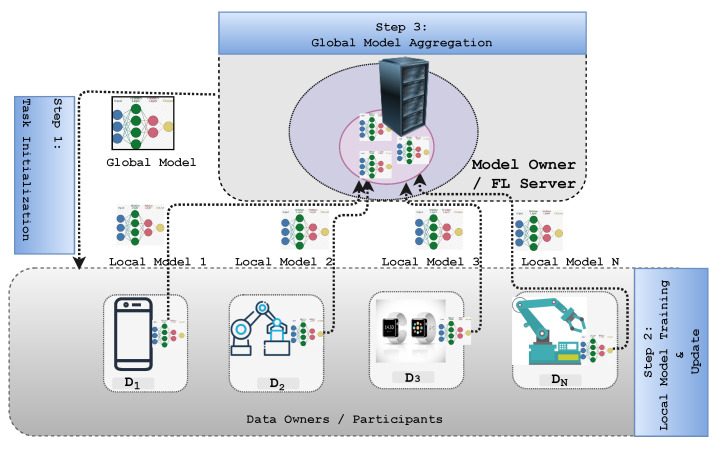
An FL general process with the server and N participants synchronizing with a server and updating it with an updated global model in a single communication round.

**Figure 4 sensors-22-00450-f004:**
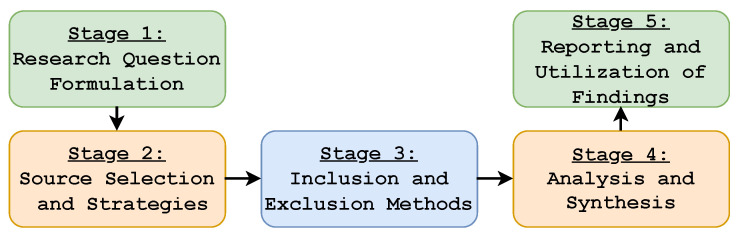
Research methodology adopted in the current study [103].

**Figure 5 sensors-22-00450-f005:**
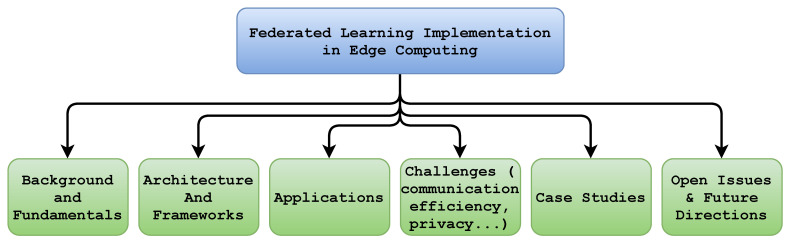
Taxonomy of federated learning in EC papers based on the high-level classification to be analyzed in this survey.

**Figure 6 sensors-22-00450-f006:**
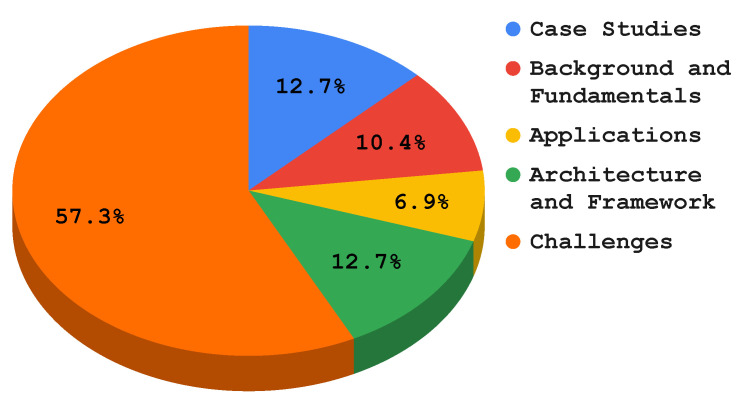
Percentage breakdown of the literature in federated learning in edge computing.

**Figure 7 sensors-22-00450-f007:**
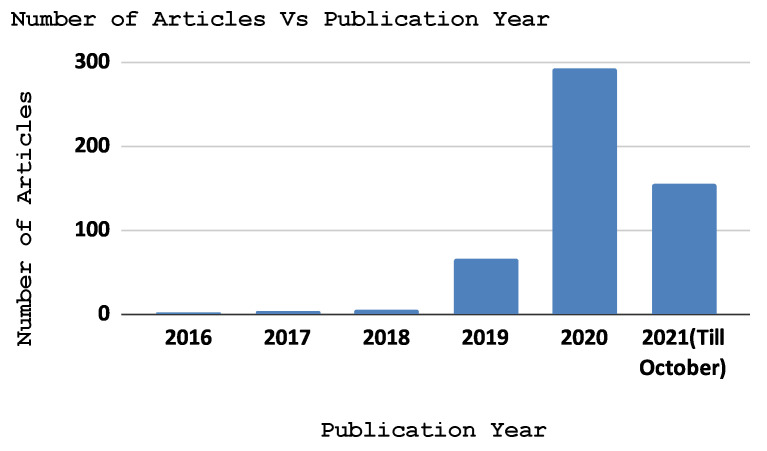
Total number of articles on federated learning in edge computing vs. publication year.

**Figure 8 sensors-22-00450-f008:**
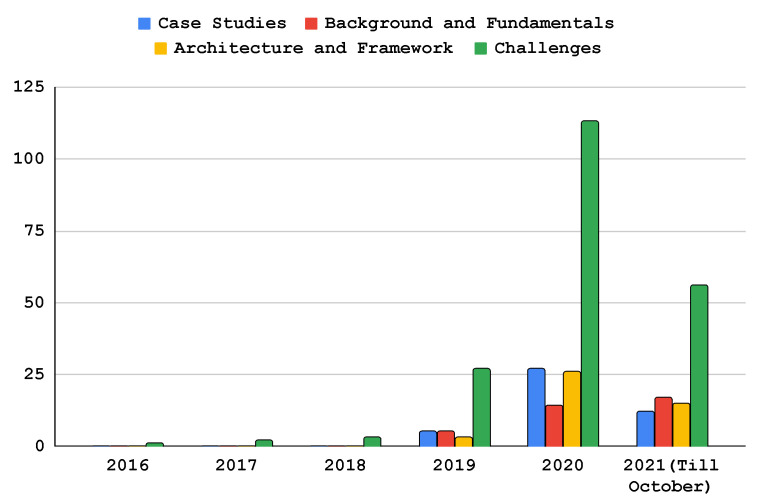
Total number of publications per year in the subject bases.

**Figure 9 sensors-22-00450-f009:**
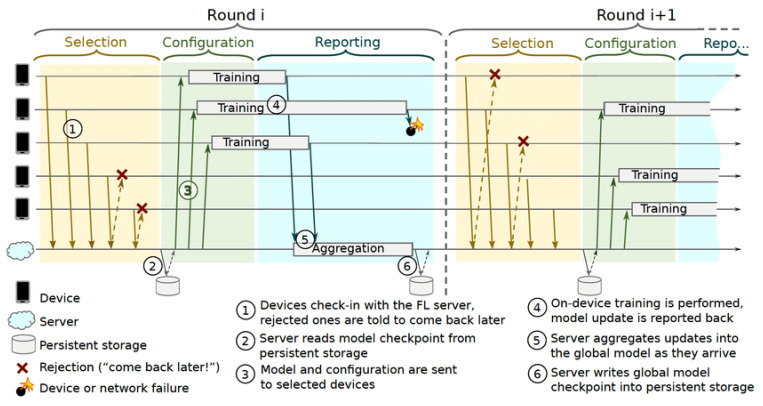
Federated learning protocol [14].

**Figure 10 sensors-22-00450-f010:**
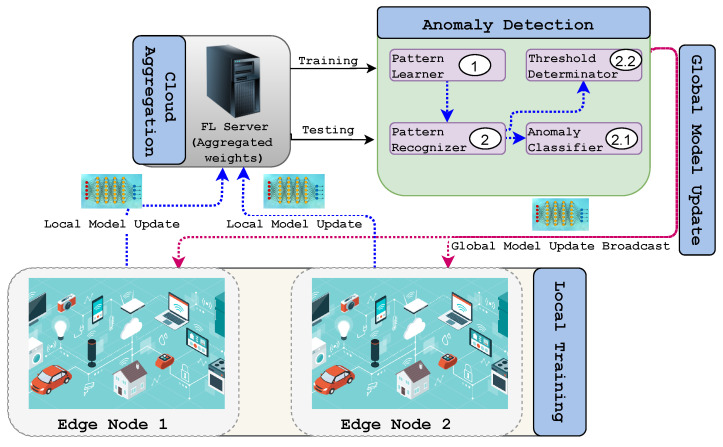
Federated learning-based anomaly/attack detection framework for edge computing.

**Figure 11 sensors-22-00450-f011:**
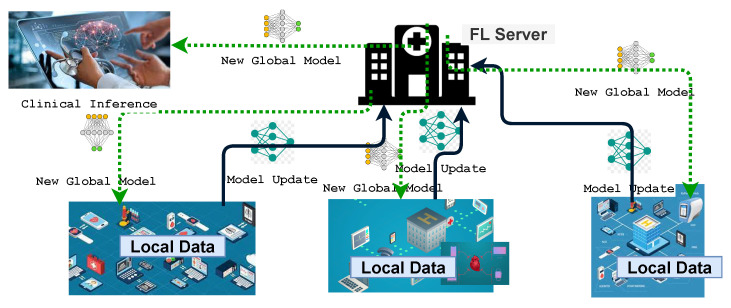
Federated learning in smart healthcare with an edge-computing-enabled infrastructure.

**Figure 12 sensors-22-00450-f012:**
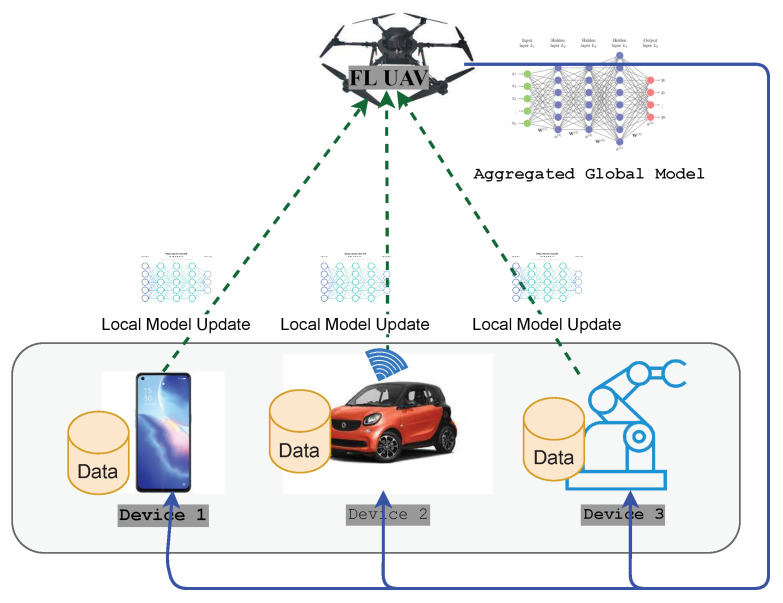
Federated Learning in UAVs with an edge-computing-enabled infrastructure.

**Table 1 sensors-22-00450-t001:** Mathematical expressions of activation functions.

Activation Function	Equation	Range
Linear Function	f(x)=x	(−∞,∞)
Step Function	$f\left(x\right)=\left\{\begin{array}{cc} 0, & \mathrm{for}\;x\;<\;0\\ 1, & \mathrm{for}\;x\geq 0 \end{array}\right.$	{0,1}
Sigmoid Function	$f\left(x\right)=\sigma \left(x\right)=\frac{1}{1+e^{-x}}$	(0,1)
Hyperbolic Tangent Function	$f\left(x\right)=\tanh\left(x\right)=\frac{e^{x}-e^{-x}}{e^{x}+e^{-x}}$	(0,1)
ReLU	$f\left(x\right)=\left\{\begin{array}{cc} 0, & \mathrm{for}\;x\;<\;0\\ x, & \mathrm{for}\;x\geq 0 \end{array}\right.$	(0,∞)
Leaky ReLU	$f\left(x\right)=\left\{\begin{array}{cc} 0.01, & \mathrm{for}\;x\;<\;0\\ x, & \mathrm{for}\;x\geq 0 \end{array}\right.$	(−∞,∞)
Swish Function	$f\left(x\right)=x\;\sigma \left(\beta \left(x\right)\right)=\left\{\begin{array}{cc} \frac{x}{2}, & \beta =0\\ max(0,x), & \beta \to \infty \end{array}\right.$	(−∞,∞)

**Table 3 sensors-22-00450-t003:** Comparison of hardware accelerators for the implementation of federated learning in edge computing.

Name	Owner	Pros	Cons
CPU/GPU	NVIDIA and Radeon	High memory, bandwidth, and throughput	Consumes a large amount of power
FPGA	Intel	High performance per watt of power consumption, reduced costs for large-scale operations, excellent choice for battery-powered devices and on cloud servers for large applications	It requires a significant amount of storage, external memory and bandwidth, and computational resources on the order of billions of operations per second
ASIC	Intel	Minimizes memory transfer, most energy efficient compared to FPGAs and GPUs, and best computational speed compared to FPGAs and GPUs	Long development cycle, Lack of flexibility to handle varying DL network designs

## Data Availability

Not applicable.

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
