# Peer review of "Federated Learning in Edge Computing: A Systematic Survey"

_sensors, 2022, doi:10.3390/s22020450_

Round 1

Reviewer 1 Report

-What is the major contribution of the work? Are the review, classification, and discussion of recent works in the area?

-A new section could be of great contribution if it would lightly describe the systematic review, as a brief introduction, use the graphs to show the trends in the area, and finally include an good, detailed description of the obtained results. This would provide a different and interesting perspective for the work.

-The authors could augment Section 7 into discussion where they present also present the more detailed challenges and future [research] directions in a more objective manner.

-Little is described or explained about figures and tables in the text. The authors should keep in mind that this manuscript is a survey, where the authors must target instructing the readers and clearly describing concepts and arguments.

-In section 6.1 the authors could cite the following works: -A low-cost smart home automation to enhance decision-making based on fog computing and computational intelligence -NodePM: A Remote Monitoring Alert System for Energy Consumption Using Probabilistic Techniques -ResiDI: Towards a smarter smart home system for decision-making using wireless sensors and actuators -A fog-enabled smart home solution for decision-making using smart objects

-I still feel that the survey lacks depth in covering the existing in the proposed area of the survey. A systematic review allows to deterministically pinpoint, list, analyze, and described the relevant works. However, the manuscript does not assign the proper attention when describing and discussing the surveyed works. 

Author Response

We would like to thank the reviewer for his/her valuable and constructive comments and feedbacks.  Below, we detail our responses and changes based on these comments and feedbacks:

Comment # 1: “What is the major contribution of the work? Are the review, classification, and discussion of recent works in the area?”

Authors response:  To illustrate the future scope of the implementation of FL in the Edge Computing (EC) paradigm, we perform a Systematic Literature Review (SLR) to compare, analyze, explore, and understand the current attempts and directions. Our contributions in this article are as follows:  1) We review the fundamentals of EC and FL, and then 2) we review the existing works related to FL in EC. In addition, 3) we describe the protocols, architecture, framework, and hardware requirements for FL implementation in the EC environment. Additionally, 4) we discuss applications, challenges, and existing solutions for edge FL. Finally, 5) we detail two relevant case studies of applying FL in EC, and 6) we identify open issues and potential directions for future research. In the article, we highlight our contributions in section 2.3 on page 11 for further information, which are highlighted in blue color for your reference.

Comment # 2: “A new section could be of great contribution if it would lightly describe the systematic review, as a brief introduction, use the graphs to show the trends in the area, and finally include a good, detailed description of the obtained results. This would provide a different and interesting perspective for the work.”  

Authors response:  Thank you for this valuable comment and suggestion, we addressed this comment in section 4: 4.1 – 4.5 on pages 11, 12, and 13 which are highlighted in blue color for your reference.

Comment # 3: “The authors could augment Section 7 into discussion where they also present the more detailed challenges and future [research] directions in a more objective manner.”

Authors response:  Thank you for this comment, we addressed this in section 7 on pages 29 and 30 which are highlighted in blue color for your reference.

Comment # 4: “Little is described or explained about figures and tables in the text. The authors should keep in mind that this manuscript is a survey, where the authors must target instructing the readers and clearly describing concepts and arguments.”

Authors response:  We considered this comment in figure 1: (page 3), Table 2:(page 11), figure 10: (pages 21 and 22) which are highlighted in blue color for your reference.

Comment # 5: “In section 6.1 the authors could cite the following works: -A low-cost smart home automation to enhance decision-making based on fog computing and computational intelligence -NodePM: A Remote Monitoring Alert System for Energy Consumption Using Probabilistic Techniques -ResiDI: Towards a smarter smart home system for decision-making using wireless sensors and actuators -A fog-enabled smart home solution for decision-making using smart objects”

Authors response: We included these references ([206] – [210]) and discussed them in section 6.1 on page 27 in the article for your reference.

Comment # 6: “I still feel that the survey lacks depth in covering the existing in the proposed area of the survey. A systematic review allows to deterministically pinpoint, list, analyze, and described the relevant works. However, the manuscript does not assign the proper attention when describing and discussing the surveyed works.” 

Authors response: We added more details in every section to address this comment, which are highlighted in blue text color for your reference.

Reviewer 2 Report

The paper presents a comprehensive review of federated learning. Different aspects  are presented, including architectures, domains where federated learning is applied, etc.

The lack of this review consists in comparing also the obtained results of different models of federated learning applied in different domains. That were described applications for healthcare and UAVs but the obtained results were not discussed, eg. what is the best approach / architecture for different situations.

Also in case of security and privacy problems, that are presented a set of methods, but it be useful to list also a comparison between presented methods (advantages, disadvantages, resource required).

After reading this review, the reader must remain with a clear idea what approaches are adequate for different situations.

Author Response

We would like to thank the reviewer for his/her valuable and constructive comments and feedbacks.  Below, we detail our responses and changes based on these comments and feedbacks:

Comment # 2: “The lack of this review consists in also comparing the obtained results of different models of federated learning applied in different domains. Those were described applications for healthcare and UAVs but the obtained results were not discussed, e.g. what is the best approach/architecture for different situations.”

Author response:  We thank you for this valuable comment, we considered it in section 6.1 on pages 27 and 28, and section 6.2 on pages 29 which are highlighted in blue color for your reference.

Comment # 3: “Also in case of security and privacy problems, that are presented a set of methods, but it is useful to list also a comparison between the presented methods (advantages, disadvantages, resource required).”

Author response: We considered your comment, and we compared the different methods addressing security and privacy in Edge-enabled FL in section 5.4.3 on pages 25, 26, and 29, which are highlighted in blue color for your reference.

Comment # 4: “After reading this review, the reader must remain with a clear idea of what approaches are adequate for different situations.”

Author response:  We add some details in every section to address your comment which are highlighted in blue text color for your reference

Round 2

Reviewer 1 Report

The authors adequately responded to the suggestions.

Reviewer 2 Report

Since all my comments were addressed, I recommend to publish the paper.